# Regret Bounds for Multilabel Classification in Sparse Label Regimes

**Róbert Busa-Fekete**
Google Research
busarobi@google.com

**Heejin Choi**
Google
heejinc@google.com

**Krzysztof Dembczyński**
Yahoo Research
Poznan University of Technology
kdembczynski@cs.put.poznan.pl

**Claudio Gentile**
Google Research
cgentile@google.com

**Henry W. Reeve**
University of Bristol
henry.reeve@bristol.ac.uk

**Balázs Szörényi**
Yahoo Research
szorenyibalazs@gmail.com

## Abstract

Multi-label classification (MLC) has wide practical importance, but the theoretical understanding of its statistical properties is still limited. As an attempt to fill this gap, we thoroughly study upper and lower regret bounds for two canonical MLC performance measures, Hamming loss and Precision@$\kappa$. We consider two different statistical and algorithmic settings, a non-parametric setting tackled by plug-in classifiers à la $k$-nearest neighbors, and a parametric one tackled by empirical risk minimization operating on surrogate loss functions. For both, we analyze the interplay between a natural MLC variant of the low noise assumption, widely studied in binary classification, and the label sparsity, the latter being a natural property of large-scale MLC problems. We show that those conditions are crucial in improving the bounds, but the way they are tangled is not obvious, and also different across the two settings.

## 1 Introduction

Multi-label classification (MLC) is a natural generalization of binary classification and multi-class classification. It can be seen as a composition of several binary problems for the same instances, or as a multi-class problem where more than one label can be assigned to a given instance. Several applications can be naturally modeled as a multi-label problem, including image and video annotation for multimedia search [13], tagging text documents for categorization of Wikipedia articles [9], recommendation of bid words for online ads [27], and beyond. MLC is a well-established sub-field of supervised learning, that has generated a tremendous interest, mainly of experimental nature, introducing a lot of interesting and diverse approaches (for example, [10, 26, 17, 34, 33, 19] ). They have also been described in multiple survey papers across years, e.g., [30, 4]. The main concerns in MLC research are label dependencies (marginal and conditional, see the discussion in [12]), label sparsity [18, 34], and minimization of a wide spectrum of performance metrics.

Interestingly, the theoretical understanding of MLC learning problems is still limited. This might be due to the daunting presence of multiple and diverse loss functions that can be defined over subsets of labels. Popular loss functions include Hamming loss, Precision@$\kappa$, or subset 0/1 loss. Each such MLC loss function has its pros and cons, and there is no consensus among researchers what is the right performance metric to be used [12, 20], as the choice is certainly domain specific. The theoretical results obtained so far mainly deal with the analysis of Bayes optimal classifiers for the wide variety of MLC loss functions [12, 31, 24, 25], Fisher-like consistency results [11, 16], or

36th Conference on Neural Information Processing Systems (NeurIPS 2022).

surrogate regret bounds [22, 33]. Results concerning generalization bounds have also been published in [6, 34, 32].

We fill the gap in the landscape of theoretical results by providing upper and lower finite-sample regret bounds in MLC with a focus on computational efficiency. We consider two learning setups, a non-parametric and a parametric one. Within the former, we focus specifically on a $k$-nearest neighbor ($k$-NN) classifier, because of its attractive properties in MLC settings. Beyond computational efficiency, a main reason why $k$-NN is a convenient approach is that it works "out of the box", operating only in the feature space, independent of the number of labels, hence making it a very versatile tool for MLC tasks. It has also been studied in MLC, mainly empirically, for example in [35]. In the parametric setting, we introduce a MLC counterpart to the surrogate loss regret reductions introduced in [3], thereby considering empirical risk minimization solutions applied to convex surrogate loss functions. We restrict our attention to Hamming loss and Precision@$\kappa$, and thoroughly investigate the interplay between two natural assumptions, the low noise condition and the label sparsity, in determining rates of convergence.

Our results build on a number of former contributions. The analysis of $k$-NN classifiers has been motivated by the results in [29, 28]. The proof techniques we rely upon in deriving regret bounds are related to those adopted for high probability regret bounds in binary classification [5]. To obtain lower bounds we follow to some extent results of [2], as the proof uses reduction of a sparse MLC problem to binary classification. The most similar results to ours have been published in [8, 7]. In [8], the authors analyze a modified version of Hamming loss which assigns different weights to relevant and irrelevant labels. The paper also assumes sparsity, but that assumption is stronger than ours. The guarantees obtained are in terms of generalization and regret bounds. [7] does not directly use Precision@$\kappa$, but the setting analyzed there is similar to the one for Precision@$\kappa$. That paper also formulates a margin condition similar to ours. Yet, the regret bounds contained there are not of a high probability type, but only bounds in expectation, similar to [2]. Moreover, [7] does not contain (matching) lower bounds. Our results for the parametric case build on [3], which contains a regret analysis, also under low noise condition, for surrogate losses in binary classification. Our generalization takes additionally multi-label sparsity into account.

**Main results.** We consider novel label-wise margin (low-noise) conditions that are sufficient to get fast rates. In addition, we assume that at most $s$ out of all $m$ labels can have conditional probability arbitrarily close to 1, whereas for the rest of the labels this probability is upper bounded by a small constant. Such label *sparsity* assumption is essentially motivated by real-world multi-label problems with a large number of potential labels, where only a few such labels are assigned to single instances, a paradigmatic example being image or text annotation.

In the non-parametric setting, we consider a MLC variant of $k$-NN as our method of choice, since this method is well-understood in the binary case, and can be shown to achieve fast rates for 0/1-loss (which is in fact the binary counterpart to Hamming loss). Under multi-label margin and sparsity assumptions, we present a high probability bound for the Hamming loss regret of $k$-NN which is of the form $O\big(s/m \cdot (\log(s/\delta)/n)^{(\lambda(1+\alpha))/(2\lambda+1)}\big) + \delta$, where $s$ is the sparsity parameter, $\lambda$ is the smoothness of the regression function with respect to the infinity norm $\|.\|_\infty$, $\alpha$ is the exponent in the margin condition, and $\delta$ is the desired confidence level. In the special case where no sparsity is assumed we have $s = m$, hence the bound boils down to a more "standard" upper bound for the Hamming loss regret. We would like to emphasize that applying a binary regret analysis, such as the one in [5], to each of the $m$ labels independently cannot leverage global properties like label sparsity. Thus, in order to take advantage of label sparsity, a more involved regret analysis is needed. Also observe that, by making the margin condition more favorable (that is, by increasing $\alpha$), the dependence on the sparsity ratio $s/m$ remains unaffected. In case of Precision@$\kappa$, we do not need label sparsity: a margin condition, that controls the gap between the $\kappa$-th and the $(\kappa + 1)$-th largest conditional distribution, is by itself enough to obtain regret bounds that scale gracefully with the number of labels. Under this assumption, we obtain a regret bound for the multi-label $k$-NN classifier of the form $(\log(m/\delta)/n)^{(\lambda(1+\beta))/(2\lambda+1)} + \delta$, where $\beta$ is the margin exponent for the $\kappa$-th and $(\kappa + 1)$-th largest conditional distribution.

In the parametric setting, we consider regret bounds for function classes with finite VC dimension. Under the margin and sparsity assumptions, we relate the Hamming loss regret $\text{Reg}_{\ell_H}$ of an arbitrary

MLC classifier to its MLC surrogate loss regret $\text{Reg}_{\ell}$ via bounds of the form

$$\text{Reg}_{\ell_H} = O\Big( (s/m)^{\frac{\nu-1}{\alpha+1}} \text{Reg}_{\ell}^{\frac{\alpha+1}{\alpha+\nu}} \Big),$$

where $\nu$ is a parameter related to the curvature of the surrogate loss $\ell$ (e.g., $\nu = 2$ for both truncated square and exponential losses). In turn, these bounds can be used in combination with standard uniform convergence results to control $\text{Reg}_{\ell}$, e.g.,

$$\text{Reg}_{\ell} = O\left( \frac{d + \log(m/\delta)}{\sqrt{n}} + a \right),$$

where $d$ is the pseudo-dimension of the VC class under consideration, and $a$ is its approximation error. Regret bounds for Precision@$\kappa$ are reduced to those for Hamming loss.

As a last contribution, we derive MLC regret lower bounds for our MLC setups revealing that, at least in the non-parametric case, our upper bound for Hamming loss is optimal up to a $\log s$ factor, and that our regret upper bound for Precision@$\kappa$ is optimal up to a $\log m$ factor.

## 2 Preliminaries and notation

We assume to have a feature space $\mathcal{X}$ endowed with a metric $\rho_{\mathcal{X}}$, and a set of $m$ labels. We denote by $\mathcal{Y} := \{0, 1\}^m$ the set of binary vectors encoding the set of all $2^m$ possible label combinations. We shall assume the existence of a pair of random variables $\boldsymbol{X} \in \mathcal{X}$ and $\boldsymbol{Y} \in \mathcal{Y}$ with joint distribution $P$. Let $\mathcal{M}(\mathcal{X}, \mathcal{Y})$ denote the set of all *classifiers*, which are measurable mappings $\phi : \mathcal{X} \to \mathcal{Y}$. The goal of the learner is to obtain a classifier $\phi \in \mathcal{M}(\mathcal{X}, \mathcal{Y})$ such that $\phi(\boldsymbol{x}) \approx \boldsymbol{y}$ for typical pairs $(\boldsymbol{x}, \boldsymbol{y}) \sim P$. The precise sense in which "$\phi(\boldsymbol{x}) \approx \boldsymbol{y}$" is provided by a loss function $\ell : \mathcal{Y} \times \mathcal{Y} \to [0, \infty)$, with the interpretation that $\ell(\tilde{\boldsymbol{y}}, \boldsymbol{y})$ measures the cost incurred when the learner predicts $\tilde{\boldsymbol{y}}$ while the true (multi-)label is $\boldsymbol{y}$.

The problem of multi-label classification can be defined as finding a (vector-valued) *classifier* $\phi(\boldsymbol{x}) = (\phi_j(\boldsymbol{x}))_{j \in [m]}$, that minimizes the *expected loss* (or *risk*):

$$R_{\ell}(\phi) = \mathbf{E}\left[ \ell(\phi(\boldsymbol{X}), \boldsymbol{Y}) \right] ,$$

the expectation being over $\boldsymbol{X} \times \boldsymbol{Y} \sim P$. For a given loss function $\ell$, the optimal classifier, often called the *Bayes (optimal) classifier* is defined as $\phi_{\ell}^* = \text{argmin}_{\phi \in \mathcal{M}(\mathcal{X}, \mathcal{Y})} R_{\ell}(\phi)$. When $\ell$ is clear from the surrounding context, we shall omit the subscript and simply write $\phi^*$. We can measure the statistical performance of a given classifier $\phi$ in terms of its *regret*, henceforth denoted by $\text{Reg}_{\ell}(\phi)$, quantifying the sub-optimality of $\phi$. Specifically, $\text{Reg}_{\ell}(\phi)$ measures the extent to which the expected loss of $\phi$ exceeds the expected loss of the Bayes classifier:

$$\text{Reg}_{\ell}(\phi) := R_{\ell}(\phi) - R_{\ell}(\phi_{\ell}^*) . \tag{1}$$

From a purely statistical standpoint, the goal of learning is then finding a classifier $\phi$ with regret as small as possible, ideally equal to zero.

Computing $\phi^*$ requires prior knowledge of the data distribution $P$. Yet, $P$ is assumed to be unknown and accessible only through a dataset (or *statistical sample*) $\mathcal{D} = \{(\boldsymbol{X}^i, \boldsymbol{Y}^i)\}_{i \in [n]}$, where the pairs $(\boldsymbol{X}^i, \boldsymbol{Y}^i)$ are drawn i.i.d. from $P$. In what follows, we denote by $\mu$ the marginal distribution of $P$ over $\boldsymbol{X}$, so that $\mu(A) = P(\boldsymbol{X} \in A)$. Moreover, we define the multi-label regression function $\eta : \mathcal{X} \to [0, 1]^m$ by $\eta(\boldsymbol{x}) = (\eta_j(\boldsymbol{x}))_{j \in [m]}$, whose $j$-th component $\eta_j(\boldsymbol{x})$ is the conditional probability of the $j$-th label being one: $\eta_j(\boldsymbol{x}) = \mathbf{P}(Y_j = 1 | \boldsymbol{X} = \boldsymbol{x})$.

## 3 Losses for multi-label classification

We focus on two widely used loss functions for MLC, namely Hamming loss and Precision@$\kappa$, which we now define. The Hamming loss $\ell_H : \mathcal{Y} \times \mathcal{Y} \to [0, 1]$ is defined for pairs $\boldsymbol{y} = (y_j)_{j \in [m]} \in \mathcal{Y}$ and $\hat{\boldsymbol{y}} = (\hat{y}_j)_{j \in [m]} \in \mathcal{Y}$ by

$$\ell_H(\hat{\boldsymbol{y}}, \boldsymbol{y}) := \frac{1}{m} \sum_{j \in [m]} \mathbb{I}\{\hat{y}_j \neq y_j\} .$$

Hence, $\ell_H(\hat{y}, y)$ simply counts the fraction of mispredicted labels. We observe that for all $\phi \in \mathcal{M}(\mathcal{X}, \mathcal{Y})$ we have

$$R_{\ell_H}(\phi) = \frac{1}{m} \sum_{j \in [m]} \int \Big( \phi_j(\boldsymbol{x})\,(1 - \eta_j(\boldsymbol{x})) + (1 - \phi_j(\boldsymbol{x}))\,\eta_j(\boldsymbol{x}) \Big) d\mu(\boldsymbol{x}) \,,$$

implying that $\phi^*_{\ell_H} : \mathcal{X} \to \mathcal{Y}$, defined by

$$\phi^*_{\ell_H}(\boldsymbol{x}) = (\mathbb{I}\{\eta_j(\boldsymbol{x}) \geq 1/2\})_{j \in [m]}$$

is the Bayes optimal classifier for $\ell_H$. Moreover, it is not hard to see that for all $\phi \in \mathcal{M}(\mathcal{X}, \mathcal{Y})$ we have

$$\mathrm{Reg}_{\ell_H}(\phi) = \frac{1}{m} \sum_{j \in [m]} \int \mathbb{I}\{\phi_j(\boldsymbol{x}) \neq (\phi^*_{\ell_H}(\boldsymbol{x}))_j\} \times |2\eta_j(\boldsymbol{x}) - 1|\, d\mu(\boldsymbol{x}) \,.$$

To define the Precision@$\kappa$ loss, we need to restrict the predicted vectors $\hat{y}$ to lie in $\mathcal{Y}_\kappa = \{y \in \mathcal{Y} : \|y\|_1 = \kappa\}$. Then we have[1]

$$\ell_{@\kappa}(\hat{y}, y) = 1 - \frac{1}{\kappa} \sum_{j \in T(\hat{y})} \mathbb{I}\{y_j = 1\} \,, \qquad (2)$$

where $T(\hat{y}) = \{j \in [m] : \hat{y}_j = 1\}$ is the set of labels output by the predictor (corresponding to the labels of value one). Let $\phi^\kappa(f) : R^m \to \mathcal{Y}_\kappa$ be a mapping from a vector of real-valued scores to a binary vector with $\kappa$ ones. Then, the corresponding risk is:

$$R_{\ell_{@\kappa}}(\phi^\kappa(f)) = 1 - \frac{1}{\kappa} \int \sum_{j \in T(\phi^\kappa(f(\boldsymbol{x})))} \eta_j(\boldsymbol{x}) d\mu(\boldsymbol{x}) \,.$$

Based on this, one can obtain the Bayes optimal classifier $\phi^*_{\ell_{@\kappa}}$ for $\ell_{@\kappa}$ as one that predicts on the given $\boldsymbol{x}$ in input the $\kappa$ classes with the highest conditional probabilities $\eta_j(\boldsymbol{x})$ as positives, i.e

$$\phi^*_{\ell_{@\kappa}}(\boldsymbol{x}) := \big(\mathbb{I}\{i \in T^\kappa(\eta(\boldsymbol{x}))\}\big)_{i \in [m]} \,,$$

where $T^\kappa : \mathbb{R}^m \to [m]^\kappa$ returns the set of $\kappa$ indices that are largest for a real-valued input vector. The regret of any classifier $\phi$ can be written as

$$\mathrm{Reg}_{\ell_{@\kappa}}(\phi) = \frac{1}{\kappa} \int \Big( \sum_{i \in T^\kappa(\eta(\boldsymbol{x}))} \eta_i(\boldsymbol{x}) - \sum_{j \in T^\kappa(\phi(\boldsymbol{x}))} \eta_j(\boldsymbol{x}) \Big) d\mu(\boldsymbol{x}) \,.$$

## 4 Sparsity and margin condition

We introduce two assumptions which are fairly natural for MLC scenarios. The first one encodes the intuition that in typical MLC settings, when the total number of classes/labels $m$ is large, only very few labels are active for any given region in the feature space, that is, each item $\boldsymbol{x}$ comes with a *sparse* label vector $\boldsymbol{y}$.

**Assumption 4.1** (Sparsity assumption). Define $A(s, t)$ as the set of $t$-approximately $s$-sparse vectors as follows:

$$A(s, t) := \{\boldsymbol{u} + \boldsymbol{v} : \boldsymbol{u}, \boldsymbol{v} \in \mathbb{R}^m, \ \|\boldsymbol{u}\|_0 \leq s, \|\boldsymbol{v}\|_\infty \leq t\} \,.$$

Take $t \in [0, 1/2]$ and $s \in \mathbb{N}$. We shall say that the $t$-approximate $s$-sparsity assumption holds if the multi-label regression function $\eta(\boldsymbol{x}) \in A(s, t)$ for all $\boldsymbol{x} \in \mathcal{X}$.

Assumption 4.1 always holds with $s = m$ and $t = 0$, but for many distributions encountered in practice we expect it to hold with a much smaller value of $s$, possibly coupled with $t > 0$. In fact, this assumption generalizes the standard sparsity assumption of "at most $s$ relevant labels per instance" by adding a term for a long tail. Yet, it does not introduce a hard constraint, since it applies to

---

[1]In the relevant literature on the subject, Precision@$\kappa$ is often regarded as a utility function defined as Precision@$\kappa = 1 - \ell_{@\kappa}$. To be consistent with our discussion, we instead define Precision@$\kappa$ as a loss function.

$\eta(\boldsymbol{x})$. According to this sparsity assumption, the expected number of labels per instance $\boldsymbol{x}$ is at most $\mathbf{E}[\sum_{j=1}^{m} Y_j | \boldsymbol{X} = \boldsymbol{x}] \leq s + (m - s) \cdot t$. On real world multi-label datasets with large label spaces, the expected number of labels is usually less than 100, while the number of possible labels $m$ can be three or four orders of magnitude bigger [1, 21, 27].

The next assumption is a MLC variant of Tsybakov's margin (or low noise) assumption (e.g., [2]). It ties up "noise" present on different labels to occur simultaneously on those labels.

**Assumption 4.2** (Margin assumption). Let $\alpha \geq 0$ and $C_\alpha \geq 1$ be constants. We shall say that the margin assumption holds with exponent $\alpha$ and constant $C_\alpha$ if the following holds for all $\epsilon \in (0, 1]$:

$$\mu\left( \bigcup_{j \in [m]} \{ \boldsymbol{x} \in \mathcal{X} : 0 < |2\eta_j(\boldsymbol{x}) - 1| \leq \epsilon \} \right) \leq C_\alpha \cdot \epsilon^\alpha .$$

## 5   Non-parametric setting

### 5.1   The $k$-NN regression estimator

Given $\boldsymbol{x} \in \mathcal{X}$, we let $\{\tau_{n,q}(\boldsymbol{x})\}_{q \in [n]}$ be an enumeration of $[n]$ such that for each $q \in [n-1]$,

$$\rho_{\mathcal{X}}\left( \boldsymbol{x}, \boldsymbol{X}^{\tau_{n,q}(\boldsymbol{x})} \right) \leq \rho_{\mathcal{X}}\left( \boldsymbol{x}, \boldsymbol{X}^{\tau_{n,q+1}(\boldsymbol{x})} \right) .$$

In words, for given $\boldsymbol{x} \in \mathcal{X}$, $\{\tau_{n,i}(\boldsymbol{x})\}_{i \in [n]}$ sorts $\{\boldsymbol{X}^1, \ldots, \boldsymbol{X}^n\}$ in increasing order of $\rho_{\mathcal{X}}$-distance to $\boldsymbol{x}$.

The $k$-nearest neighbor regression estimator $\hat{\eta}^{n,k} : \mathcal{X} \to [0,1]^m$ is given by $\hat{\eta}^{n,k}(\boldsymbol{x}) := (\hat{\eta}_j^{n,k}(\boldsymbol{x}))_{j \in [m]}$ where

$$\hat{\eta}_j^{n,k}(\boldsymbol{x}) := \frac{1}{k} \cdot \sum_{i \in [k]} Y_j^{\tau_{n,i}(\boldsymbol{x})}. \tag{3}$$

The plug-in multi-label classifier for Hamming loss that is based on the $k$-NN regression function $\hat{\eta}^{n,k}$ is denoted by $\phi_{\ell_H}^{n,k}$, and defined as

$$\phi_{\ell_H}^{n,k}(\boldsymbol{x}) = \left( \mathbb{I}\{\hat{\eta}_j^{n,k}(\boldsymbol{x}) \geq 1/2\} \right)_{j \in [m]} .$$

Similarly, the plug-in multi-label classifier for Precision@$\kappa$ loss that is based on the $k$-NN regression function is denoted by $\phi_{\ell_{@\kappa}}^{n,k}$ and defined as

$$\phi_{\ell_{@\kappa}}^{n,k}(\boldsymbol{x}) := \left( \mathbb{I}\{j \in T^\kappa(\hat{\eta}_j^{n,k}(\boldsymbol{x}))\} \right)_{j \in [m]} .$$

In this paper, we shall scrutinize the performance of these $k$-NN classifiers in terms of regret under some assumptions on the classification problem at hand.

Given $\boldsymbol{x} \in \mathcal{X}$ and $r > 0$, let $B_r(\boldsymbol{x})$ denote the open metric ball of radius $r$, centered at $\boldsymbol{x}$.

**Assumption 5.1** (Smoothness assumption). Suppose we have an exponent $\lambda \in (0, 1]$, a constant $C_\lambda > 0$ and a semi-norm $\| \cdot \|$ on $\mathbb{R}^m$. We shall say that the measure-smoothness assumption holds with parameters $\lambda, C_\lambda, \| \cdot \|$ if the following holds for all $\boldsymbol{x}_0, \boldsymbol{x}_1 \in \mathcal{X}$:

$$\|\eta(\boldsymbol{x}_0) - \eta(\boldsymbol{x}_1)\| \leq C_\lambda \cdot \mu\left( B_{\rho_{\mathcal{X}}(\boldsymbol{x}_0, \boldsymbol{x}_1)}(\boldsymbol{x}_0) \right)^\lambda .$$

This kind of assumption is typical for $k$-NN. It says that the regression function $\eta(\boldsymbol{x})$ does not change much in the neighborhood of every input $\boldsymbol{x}$. This is essentially needed for $k$-NN convergence. The specific version adopted here is from [5], and was further employed, e.g., in [29]. Note that, more generally, a universal rate of convergence cannot be established without assuming regularity properties of the data distribution, such as smoothness via absolute continuity. Results of that kind are beyond the scope of this study. We refer the reader to [14] for details on $L_1$ consistency and its connection to the rate of convergence. In this paper, we end up using Assumption 5.1 with $|| \cdot ||$ being the infinity norm, not an arbitrary semi-norm.

## 5.2 Regret upper bound for Hamming loss

This multi-label margin assumption is a sufficient condition for fast rate $O(1/n)$. Its binary classification counterpart is well-understood in the binary classification literature.

We have the following high probability upper bound for the regret in terms of Hamming loss for the $k$-NN classifier $\phi_{\ell_H}^{n,k}$, which is the main result of this section.[2]

**Theorem 5.2.** *Let Assumption 5.1 hold with parameters $\lambda$, $C_\lambda > 0$, $\|\cdot\|_\infty$, Assumption 4.1 hold with parameters $s \in \mathbb{N}$, $t \in [0, 1/2)$ and Assumption 4.2 hold with parameters $\alpha$, $C_\alpha > 0$. Assume further that $k$ in $k$-NN satisfies $16(1-2t)^{-2}\log(2m/\delta) \leq k \leq ((1-2t)/(8C_\lambda))^{1/\lambda}(n/2)$. Then, with probability at least $1 - \delta$ over $\mathcal{D}$, the regret of the $k$-NN plug-in classifier $\phi_{\ell_H}^{n,k}$ obtained from (3) satisfies*

$$Reg_{\ell_H}(\phi_{\ell_H}^{n,k}) \leq \frac{2s}{m} \cdot C_\alpha \left(\Delta(n,k,s,\delta)\right)^{1+\alpha} + \delta\,,$$

*where*

$$\Delta(n,k,s,\delta) = C_\lambda \cdot \left(\frac{2k}{n}\right)^\lambda + \sqrt{\frac{\log(s/\delta)}{k}}\,.$$

*In particular with $k = \Omega\left(\log(8s/\delta)^{\lambda/(2\lambda+1)} \cdot n^{2\lambda/(2\lambda+1)}\right)$ and*

$$n = \Omega\left((\log(2m/\delta)/(1-2t)^2)^{(2\lambda+1)/2\lambda}\right)$$

*we have*

$$Reg_{\ell_H}(\phi_{\ell_H}^{n,k}) = O\left(\frac{s}{m} \cdot \left(\frac{\log(4s/\delta)}{n}\right)^{\frac{\lambda(1+\alpha)}{2\lambda+1}}\right) + \delta\,.$$

A few remarks are in order at this point.

*Remark* 5.3. (On the assumptions) The regret bound given in Theorem 5.2 requires three assumptions. The smoothness assumption of the regression function (Assumption 5.1) is natural in the sense that a universal rate of convergence cannot be established without assuming regularity properties of the data distribution (e.g., [14, 15]). Analyzing the impact of different smoothness criteria is beyond the scope of this study. Assumption 4.1 ensures label sparsity which allows $k$-NN learning to achieve regret of order $s/m$. Again, this assumption is also well motivated by practice, since most of the MLC task of practical relevance have sparse labels. Finally, Assumption 4.2 ensures fast rates, i.e., faster than $1/\sqrt{n}$. This assumption is commonly used and well-understood in the binary case. Here, we consider a natural extension to the multi-label setup which quantifies the probability of observing instances whose conditional probabilities are close to $1/2$.

*Remark* 5.4. (On MLC bounds based on binary classification) Theorem 5.2 with $s = m$ results in a high probability upper bound of the form $O\left((\log(m/\delta)/n)^{\frac{\lambda(1+\alpha)}{2\lambda+1}}\right) + \delta$. This bound coincides with the fast rate upper bound which we would achieve had we applied the regret upper bound for the binary case (e.g., Theorem 4 in [5]) to each conditional $\eta_j(\boldsymbol{x})$ independently, after setting $\delta/m$ and taking a union bound. Notice that the binary result can be applied to our setup to each $\eta_j(\boldsymbol{x})$ individually because our smoothness condition is based on $\|.\|_\infty$, so that the smoothness definition for the binary regression function in [5] readily applies to our MLC setting. Yet, we would like to stress that this need not imply that the bound so obtained is optimal. Rather, this way of viewing MLC as a set of independent binary problems sheds light on how to construct a set of problem instances on which we can prove *lower* bounds based on the binary result (see Section 5.4 for details). On the other hand, the sparsity condition $s < m$ imposes global constraints on the MLC task that makes the bounds contained in Theorem 5.2 not readily obtainable by simply reducing to $m$ independent binary problems.

*Remark* 5.5. (On fast and ultra fast rates) When $\lambda\alpha > 1/2$, the exponent of $(1/n)\log(4s/\delta)$ in the bound of Theorem 5.2 becomes larger than 1/2, giving a fast rate guarantee. Similarly, when $\lambda(\alpha - 1) > 1$, the exponent becomes larger than 1, and we obtain ultra fast rates.

---

[2]Due to space limitations, all proofs are in the appendices.

## 5.3 Margin condition for Precision@$\kappa$

The margin condition for Hamming loss ensures a gap of conditional probabilities around $1/2$ with a certain probability, which is motivated by the fact that those are the hard-to-classify instances when the target loss is the Hamming loss. Starting from this observation, one can come up with a margin condition for Precision@$\kappa$ of a similar flavor. Let us denote the $i$th largest margin given $\boldsymbol{x}$ by $\eta_{(i)}(\boldsymbol{x})$. We assume there is a gap with some quantifiable probability between the $\kappa$-th and $(\kappa+1)$-th conditional distribution, as specified next.

**Assumption 5.6** (Margin condition for Precision@$\kappa$). There exist $\beta \geq 0$ and $K_\beta \geq 0$ such that for all $\xi > 0$ we have
$$\mathbf{P}\left(\eta_{(\kappa)}(\boldsymbol{X}) - \eta_{(\kappa+1)}(\boldsymbol{X}) < \xi\right) \leq K_\beta \xi^\beta.$$

This assumption bears close resemblance to the margin condition introduced in [2] but, clearly enough, in order to make the optimal decision for Precision@$\kappa$, the classifier needs to distinguish between $\eta_{(\kappa)}(\boldsymbol{x})$ and $\eta_{(\kappa+1)}(\boldsymbol{x})$ with a certain quantifiable probability over $\boldsymbol{X}$. With this assumption at hand, one can show the following high probability regret bound for $\phi_{\ell_{@\kappa}}^{n,k}$.

**Theorem 5.7.** *Let Assumption 5.1 hold with parameters $\lambda$, $C_\lambda > 0$, $\|\cdot\|_\infty$, and Assumption 5.6 hold with parameter $\beta > 0$ and $K_\beta$. Suppose further that $k$ satisfies $4\log(2/\delta) \leq k \leq n/2$. Then, with probability at least $1 - \delta$ over $\mathcal{D}$ it holds that*
$$Reg_{\ell_{@\kappa}}(\phi_{\ell_{@\kappa}}^{n,k}) \leq K_\beta(2\Delta(n,k,\delta))^{1+\beta} + 2\delta ,$$

*where $\Delta(n,k,\delta) = C_\lambda \left(\frac{2k}{n}\right)^\lambda + \sqrt{\frac{\log(3m/\delta)}{k}}$. In particular with*
$$k = \Omega(\log(3m/\delta)^{\lambda/(2\lambda+1)} \cdot n^{2\lambda/(2\lambda+1)}),$$

*we have*
$$Reg_{\ell_{@\kappa}}(\phi_{\ell_{@\kappa}}^{n,k}) \in O\left(\left(\frac{\log(m/\delta)}{n}\right)^{\frac{\lambda(1+\beta)}{2\lambda+1}}\right) + 2\delta .$$

*Remark* 5.8. Theorem 5.7 does not require label sparsity in order to achieve a logarithmic dependence on the number of labels $m$. Label sparsity given in Assumption 4.1 does not help in general in optimizing Precision@$\kappa$, since when all conditionals are smaller than $t$, sparsity does not help find the labels with the top-$\kappa$ marginals, which is needed to make an optimal decision for Precision@$\kappa$.

*Remark* 5.9. The margin condition in Assumption 5.6 is generally inevitable in order to obtain fast rates for Precision@$\kappa$. To see this, one should delve into the proof of the lower bound in Theorem 5.11 (Appendix D), where it is shown that, for the multi-label problem constructed therein, the margin condition in Assumption 4.2 implies the one in Assumption 5.6 with $\beta = \alpha$.

## 5.4 Regret lower bounds

In this section, we present lower bounds for MLC problems with Hamming loss and Precision@$\kappa$ that scale with the label sparsity. In order to do so, we introduce two reductions that map binary data to multi-label data and multi-label classifiers to binary classifiers. These reductions are designed so as to preserve regret (scaled with sparsity for Hamming loss), in that lower bounds for binary classification turn to lower bounds for the multi-label problem with Hamming loss and Precision@$\kappa$.

**Theorem 5.10.** *Let us denote by $\mathcal{P}$ the class of probability distributions over $\mathcal{X} \times \mathcal{Y}$ for which the following assumptions hold: 1) Assumption 5.1 holds with some parameters $\lambda$, $C_\lambda > 0$, $\|\cdot\|_\infty$; 2) Assumption 4.1 holds with parameters $s \in \mathbb{N}$, $t \in [0, 1/2)$; 3) Assumption 4.2 holds with some parameters $\alpha$, $C_\alpha > 0$. Then there exists a constant $C > 0$ such that for any $n \geq 1$ and any mapping $\hat{f}_n : (\mathcal{X} \times \mathcal{Y})^n \mapsto \mathcal{M}(\mathcal{X}, \mathcal{Y})$ it holds that*
$$\sup_{P \in \mathcal{P}} \mathbf{E}\left[Reg_{\ell_H}(\hat{f}_n)\right] \geq C\mathfrak{s}n^{-\frac{\lambda(1+\alpha)}{2\lambda+1}} ,$$

*where $\mathfrak{s}$ denotes the sparsity ratio $\mathfrak{s} = s/m$.*

Lower bound for Precision@$\kappa$ can be also obtained as follows.

**Theorem 5.11.** *Let us denote by $\mathcal{P}$ the class of probability distributions over $\mathcal{X} \times \mathcal{Y}$ for which the following assumptions hold: 1) Assumption 5.1 holds with some parameters $\lambda$, $C_\lambda > 0$, $\|\cdot\|_\infty$; 2) Assumption 5.6 holds with some parameters $\beta \geq 0$ and $K_\beta \geq 0$. Then there exists a constant $C > 0$ such that for any $n \geq 1$ and any mapping $\hat{f}_n : (\mathcal{X} \times \mathcal{Y})^n \mapsto \mathcal{M}(\mathcal{X}, \mathcal{Y})$ it holds that*

$$\sup_{P \in \mathcal{P}} \mathbf{E}\left[ Reg_{\ell_{@\kappa}}(\hat{f}_n) \right] \geq C n^{-\frac{\lambda(1+\beta)}{2\lambda+1}} \ .$$

Notice that some bound $r(\delta)$ on the regret that holds with probability $(1 - \delta)$, translates into a bound $r(\delta) + \delta$ on the expected regret in a straightforward manner. Therefore any lower bound on the expected regret also applies when considering high probability bounds. As a consequence, the Hamming loss upper bound in Theorem 5.2 is optimal up to a $\log s/\delta$ factor and the regret bound for Precision@$\kappa$ given in Theorem 5.7 is optimal up to a $\log m/\delta$ factor. It is an interesting open question whether these log factors can be eliminated to get matching regret bound.

# 6 Parametric Setting

In this section, we work with multi-label scoring functions in the form of $f : \mathcal{X} \mapsto \mathbb{R}^m$. A multi-label scoring function is mapped to a multi-label classifier as $\phi_f(\boldsymbol{x}) = (\text{sgn}(f_j(\boldsymbol{x})))_{j \in [m]}$. Notice that, for convenience, we now view the multi-label space as $\{\pm 1\}^m$, instead of $\{0, 1\}^m$. Likewise the Hamming loss and Precision@$\kappa$ will be defined on $\{\pm 1\}^m$. We will work with convex surrogate loss functions with a single argument that applies to a multi-label score function as $(\ell(y_i f(\boldsymbol{x})))_{j \in [m]}$, resulting in a vector of losses for $\boldsymbol{x}$. The risk of a scoring function $f$ in terms of surrogate loss $\ell$ is then defined as $R_\ell(f) = 1/m \sum_{j=1}^m \mathbf{E}\ell(\boldsymbol{Y}_j f_j(\boldsymbol{X}))$, and the Bayes $\ell$-risk as $R_\ell^* = \inf_f R_\ell(f)$, the infimum being taken over all measurable functions. The $\ell$-regret of $f$ is as usual $\text{Reg}_\ell(f) = R_\ell(f) - R_\ell^*$. In addition, we will assume that the loss functions $\ell$ are classification calibrated, as recalled in the next definition.

**Definition 6.1.** A loss function $\ell : \mathbb{R} \mapsto \mathbb{R}$ is classification calibrated if for all $\eta \in [0, 1]$ such that $\eta \neq 1/2$ it holds that

$$\inf_{z \in \mathbb{R} \,:\, z(2\eta-1) \leq 0} \eta\ell(z) + (1 - \eta)\ell(-z) > \inf_{z \in \mathbb{R}} \eta\ell(z) + (1 - \eta)\ell(-z) \ .$$

In words, a loss function $\ell$ is classification-calibrated if the sign of the argmin classifier always equals the sign of the Bayes optimal classifier according to true underlying distribution. Examples of classification-calibrated losses are the exponential loss, the (truncated) squared loss, and the hinge loss.

Our analysis of the regret of Hamming loss and Precision@$\kappa$ extends the work of [3] to MLC. In [3], it is shown that for each classification-calibrated loss, there exists a nondecreasing function $\psi : [0, 1] \mapsto [0, \infty)$ that provides an upper bound on the regret of the 0-1 loss in terms of the regret of the surrogate loss, i.e., $\psi(\text{Reg}_{0\text{-}1}(f)) \leq \text{Reg}_\ell(f)$ for all measurable function $f : \mathcal{X} \mapsto \mathbb{R}$. We extend their 0-1 loss result to Hamming loss and Precision@$\kappa$ under sparsity assumptions. Further results, including consistency, are contained in Appendix E.

## 6.1 Regret upper bound for Hamming loss

We now give regret upper bounds for Hamming loss using classification calibrated losses. These upper bounds scale with the sparsity factor $\mathfrak{s} = s/m$, and apply to multi-label classifiers whose regret is small enough. Notice that placing small regret constraints on the multi-label classifier is unavoidable, since a classifier that has very poor performance, e.g. $f_j^{\text{bad}}(\boldsymbol{x}) = -\eta_j(\boldsymbol{x})$ for all $j$, will always suffer a regret that does not depend on $\mathfrak{s}$. Observe that a similar assumption on the performance of the classifiers is also implicitly made in Theorem 5.2 in the form of a lower bound $\Omega(\log(2m/\delta)/(1 - 2t)^2)$ on the sample size $n$, which ensures that the plug-in estimator cannot have arbitrarily large regret.

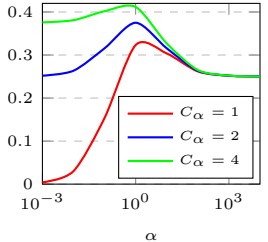

Figure 1: Maximum value of $t$ in order for Theorem 6.2 to hold. If $t$ is below the curve, an $s$ sparse classifier satisfies the condition in Theorem 6.2. Here, $\alpha$ and $C_\alpha$ are the constants in Assumption 4.1. Notice that $C_\alpha \geq 1$ when $\eta_j(\boldsymbol{x}) \neq \frac{1}{2}$ for all $\boldsymbol{x}$ and $j$.

**Theorem 6.2.** *Let Assumption 4.1 hold with parameters $s \in \mathbb{N}$, $t \in [0, 1/2)$ and Assumption 4.2 hold with parameters[3] $\alpha$ and $C_\alpha$. If $Reg_{\ell_H}(\phi_f) < \mathfrak{s}\left(2\Gamma(1-2t)\right)^{\alpha+1}$ then*

$$\Gamma\mathfrak{s}\left(\frac{Reg_{\ell_H}(\phi_f)}{\mathfrak{s}}\right)^{\frac{\alpha}{\alpha+1}}\psi\left(\frac{1}{2\Gamma}\left(\frac{Reg_{\ell_H}(\phi_f)}{\mathfrak{s}}\right)^{\frac{1}{\alpha+1}}\right) \leq Reg_\ell(f),\tag{4}$$

*where $\mathfrak{s} = \frac{s}{m}$ and $\Gamma = C_\alpha^{1/(\alpha+1)}(\alpha+1)/\alpha^{\alpha/(\alpha+1)}$.*

To see how this regret bound depends on $\mathfrak{s}$, assume $\psi(\theta)$ is of the form $\psi(\theta) = \theta^\nu$, where $\nu > 1$. This implies that $\psi$ is a strictly convex function. With this in hand, and solving for $Reg_{\ell_H}(\phi_f)$, Eq. (4) becomes

$$\text{Reg}_{\ell_H}(\phi_f) \leq \left(\Gamma^{\frac{(\nu-1)(\alpha+1)}{\alpha+\nu}}2^{\frac{\nu(\alpha+1)}{\alpha+\nu}}\mathfrak{s}^{\frac{\nu-1}{\alpha+\nu}}\text{Reg}_\ell(f)^{\frac{\alpha+1}{\alpha+\nu}}\right).\tag{5}$$

Thus the Hamming loss regret is upper bounded by a power of $Reg_\ell(f)$, multiplied by a sparsity factor of the form $\mathfrak{s}^{\frac{\nu-1}{\alpha+\nu}}$. For the sake of illustration, we can instantiate this upper bound for specific losses:

- The truncated quadratic loss $\ell(z) = (\max\{0, 1-z\})^2$ is obtained with $\nu = 2$ (see [3]). The regret in terms of this surrogate loss boils down to $O(\mathfrak{s}^{\frac{1}{\alpha+2}}\text{Reg}_\ell(f)^{\frac{\alpha+1}{\alpha+2}})$. So without margin condition, that is when $\alpha = 0$, this upper bound becomes $O(\sqrt{\mathfrak{s}\,\text{Reg}_\ell(f)})$. In contrast to Theorem 5.2, the dependency on sparsity ratio $\mathfrak{s}$ here is worse by a $\sqrt{\mathfrak{s}}$ factor. Moreover, this dependency becomes less and less relevant as $\alpha$ increases. Very similar conclusions can be drawn for the exponential loss $\ell(z) = e^{-z}$, since the corresponding $\psi(\theta) = 1 - \sqrt{1-\theta^2}$ (see again [3]) can be lower bounded by $\theta^2/2$ and upper bounded by $\theta^2$.

- Arbitrary $\nu > 1$: More generally, the exponents of the sparsity and regret factors are related to the curvature of the loss as $O(\mathfrak{s}^{1-\frac{1}{\nu}}\text{Reg}_\ell(f)^{\frac{1}{\nu}})$ when no margin condition is imposed.

*Remark 6.3.* In order to apply, Theorem 6.2 needs the regret of $\phi_f$ be below a certain threshold, i.e., $\text{Reg}_{\ell_H}(\phi_f) < \mathfrak{s}\left(2\Gamma(1-2t)\right)^{\alpha+1}$. This condition can be easily met by requiring $\phi_f$ to be an $s$ sparse classifier, and that parameter $t$ in Assumption 4.1 is not too large. Since it is usually the case that the classifiers employed in MLC are sparse, Theorem 6.2 applies to most interesting MLC problems. To see this in detail, let Assumption 4.1 hold, and consider an $s$ sparse classifier $\phi_f$. The maximum regret w.r.t. $\ell_H$ is clearly at most $2\mathfrak{s}$. Thus, when

$$\left(2\Gamma(1-2t)\right)^{\alpha+1} > 2\tag{6}$$

any $s$ sparse classifier will have regret $\text{Reg}_{\ell_H}(\phi_f)$ less then the threshold required by Theorem 6.2. Also, notice that $\Gamma = (C_\alpha)^{1/(\alpha+1)}(1+\alpha)/\alpha^{\alpha/(1+\alpha)}$ therein depends on $C_\alpha$ and $\alpha$. Therefore, for an $s$ sparse classifier, (6) is satisfied when $t$ is small enough, the maximum value of $t$ depending on $\alpha$ and $C_\alpha$, as exemplified in Figure 1.

*Remark 6.4.* In Appendix H we show that the upper bound of Theorem 6.2 cannot be improved by more than a factor of 2.

*Remark 6.5.* When working with classes $\mathcal{F}$ with finite pseudo-dimension (e.g., [15]), the regret $\text{Reg}_\ell(f)$ of any $f \in \mathcal{F}$ can as usual be decomposed into an estimation error $R_\ell(f) - R_\ell(f^*)$ and an approximation error $R_\ell(f^*) - R_\ell^*$, where $f^* \in \text{argmin}_{f\in\mathcal{F}}R_\ell(f)$. If each component $\mathcal{F}_i$ of $\mathcal{F}$ has finite pseudo-dimension $d_i$, and $\hat{f}$ is the multi-label scoring function that minimizes the

---

[3]Our notation differs from [3]. Our $\alpha$ correspond to their $\beta$.

multi-label empirical risk minimization w.r.t. the surrogate loss function, then the estimation error $R_\ell(\hat{f}) - R_\ell(f^*)$ can with probability at least $1 - \delta$ be upper bounded by an expression of the form $\sqrt{\frac{(\max_i d_i) + \log(m/\delta)}{n}}$. Combined, e.g., with (5), this yields regret upper bounds of the form

$$\mathrm{Reg}_{\ell_H}(\phi_{\hat{f}}) \leq \mathfrak{s}^{\frac{\nu-1}{\alpha+\nu}} \left( \sqrt{\frac{(\max_i d_i) + \log(m/\delta)}{n}} + R_\ell(f^*) - R_\ell^* \right)^{\frac{\alpha+1}{\alpha+\nu}} . \tag{7}$$

*Remark* 6.6. One thing to observe is that in the sparse MLC setting what is relevant is not only the dependence on the sample size $n$, but also the dependence on the sparsity factor $\mathfrak{s}$. This is because in such tasks, the total number of classes $m$ can even be larger the total number of samples $n$. Hence, in bounds like (5) or (7), it is the dependence on both $\mathfrak{s}$ and $\mathrm{Reg}_\ell(f)$ (and the interplay between the two) that better elucidates the main dependencies.

## 6.2 Regret upper bound for Precision@$\kappa$

Recall that Precision@$\kappa$ is a ranking metric that ignores the values of the scoring function components at hand, only the ranking of these values is relevant. For instance, unlike risk $R_\ell(f)$, shifting the value of $f$'s components by the same amount does not change $R_{\ell@\kappa}(T^\kappa(f))$. As a consequence, low $\mathrm{Reg}_{\ell@\kappa}(T^\kappa(f))$ does not imply low $\mathrm{Reg}_\ell(f)$. Thus, to remedy this problem, we re-center the values of the scoring function $f$ so that $f_{[\kappa]}(\boldsymbol{x}) > 0$ and $f_{[\kappa+1]}(\boldsymbol{x}) < 0$ to make it invariant to shifting values. We denote by $f'$ the centered version of $f$, i.e., $f'(x) = \left( f_j(x) - (1/2) \left( f_{[\kappa]}(x) + f_{[\kappa+1](x)} \right) \right)_{j \in [m]}$. Notice that $\mathrm{Reg}_{\ell@\kappa}(T^\kappa(f)) = \mathrm{Reg}_{\ell@\kappa}(T^\kappa(f'))$. For simplicity, we assume that $\forall \boldsymbol{x} \in \mathcal{X}$, $f_{[\kappa]}(\boldsymbol{x}) \neq f_{[\kappa+1]}(\boldsymbol{x})$ to remove ties. (One simple way to avoid ties is to add infinitesimal random noise to the values of $f$). At this point, we can rely on Theorem 6.2 to obtain an upper bound on $\mathrm{Reg}_{\ell@\kappa}(T^\kappa(f))$ in terms of the surrogate loss regret $\mathrm{Reg}_\ell(f)$, when combined with the following result.

**Theorem 6.7.** *Let $0 \leq s \leq \kappa$. If Assumption 4.1 holds with parameters $s \in \mathbb{N}$, $t \in [0, 1/2)$, then*

$$Reg_{\ell@\kappa}(T^\kappa(f)) \leq \frac{m}{2\kappa} \left( Reg_{\ell_H}(\phi_{f'}) \right) ,$$

*where $f'(x) = \left( f_j(x) - (1/2) \left( f_{[\kappa]}(x) + f_{[\kappa+1](x)} \right) \right)_{j \in [m]}$. Equivalently, for $f$ that satisfies, $\forall x$, $f_{[\kappa]}(x) > 0$, $f_{[\kappa+1]}(x) < 0$,*

$$Reg_{\ell@\kappa}(T^\kappa(f)) \leq \frac{m}{2\kappa} \left( Reg_{\ell_H}(\phi_f) \right) .$$

*The inequalities are equalities if $\forall \boldsymbol{x} \in \mathcal{X}, \forall k \in [\kappa]$, $\eta_{[k]}(\boldsymbol{x}) > \frac{1}{2}$.*

As an example, when a truncated quadratic loss is used, we obtain following upper bound by combining Theorem 6.2 with Theorem 6.7:

$$\mathrm{Reg}_{\ell@\kappa}(T^\kappa(f)) \leq \frac{m}{2\kappa} \left( \mathrm{Reg}_{\ell_H}(\phi_{f'}) \right) = O \left( \kappa^{-1} s^{\frac{1}{\alpha+2}} (m \, \mathrm{Reg}_\ell(f'))^{\frac{\alpha+1}{\alpha+2}} \right) . \tag{8}$$

In turn, $\mathrm{Reg}_\ell(f')$ in the RHS of (8) can be upper bounded as in (7).

## 7 Conclusions

The results we presented aim at filling the gap in the statistical analysis of MLC problems and methods. In the non-parametric setting, we provided upper and lower finite-sample regret bounds for $k$-NN classifiers in MLC with Hamming loss and Precision@$\kappa$ under suitable smoothness assumptions. In fact, our analysis may be extended to other loss functions, like rank loss (e.g., [11]) and micro- and macro-averaged complex performance measures (e.g., [22]). The crucial point is formulating the right assumptions for those loss functions. The general steps of these analyses will be similar to those contained in this paper, since the Bayes classifiers for these losses are in fact based on marginal quantities [23]. As an other important research avenue to explore is to make our lower bound more tight. Lower bounds that are given in this paper are tight up to log factors and to weed out this log factor seems challenging in the multi-label setup.

In the parametric setting, we considered general regret reductions from MLC surrogate losses to Hamming loss and Precision@$\kappa$, and then we presented regret bounds one can achieve by empirical risk minimization operating on the surrogate losses. One natural question is to understand to what extent these regret bounds (like those contained in Eqs. (7) and (8)) are tight, specifically when the approximation error $R_\ell(f^*) - R_\ell^*$ is strictly positive.

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
