# Supplementary material for "Regret Bounds for Multilabel Classification in Sparse Label Regimes"

This appendix contains all proofs of the results mentioned in the main body of the paper, plus further results which have been omitted there due to space limits.

## A  Technical lemmas for $k$-NN

We recall the following lemma which upper bounds the probability measure of the ball around a point $\boldsymbol{x} \in \mathcal{X}$ that contains its $k$th nearest neighbors. The proof immediately follows from the multiplicative Chernoff bound (see, e.g., Lemma 3.2 in [28]).

**Lemma A.1.** *Given $\boldsymbol{x} \in \mathcal{X}, k \in [n]$ with $k \geq 4\log(1/\delta)$, with probability at least $1 - \delta$ over $\mathcal{D} = \{\boldsymbol{X}^i\}_{i \in [n]}$ we have $\mu\big(B_{\rho(\boldsymbol{x}, \boldsymbol{X}^{\tau_{n,k}(\boldsymbol{x})})}(\boldsymbol{x})\big) \leq \frac{2k}{n}$ .*

When combined with Assumption 5.1 we obtain the following corollary.

**Corollary A.2.** *Suppose that the measure-smoothness assumption (Assumption 5.1) holds with parameters $\lambda$, $C_\lambda$, $\|\cdot\|$. Then for all $\boldsymbol{x} \in \mathcal{X}, k \in [n]$ with $k \geq 4\log(1/\delta)$, with probability at least $1 - \delta$ over $\mathcal{D}$, the following holds for all $q \in [k]$:*

$$\big\|\eta(\boldsymbol{X}^{\tau_{n,q}(\boldsymbol{x})}) - \eta(\boldsymbol{x})\big\| \leq C_\lambda \cdot (2k/n)^\lambda \ .$$

Although Corollary A.2 holds for any semi-norm, in this paper we use it with $\|.\|_\infty$. With Corollary A.2 handy, we can continue by considering in turn the two specific losses of our interest: Hamming loss (Theorem 5.2, Appendix B), and Precision@$\kappa$ (Theorem 5.7, Appendix C).

## B  Proof of Theorem 5.2

The proof of Theorem 5.2 is split into a series of technical lemmas. The proof idea can be summarized in four steps:

- As a first step, we show that the $k$-NN regression estimator $\hat{\eta}_{n,k}^j(\boldsymbol{x})$ has an error of $O(\log(s/\delta))$ for the top-$s$ labels $T^s(\eta(\boldsymbol{x}))$ for a given $\boldsymbol{x} \in \mathcal{X}$ with probability at least $1 - \delta$. This result is in Lemma B.1.

- Next, we will make use of the sparsity assumption defined in Assumption 4.1 by showing that if $k$ is big enough, i.e. $k = \Omega(\log m)$, the $k$-NN regression function $\hat{\eta}_j^{n,k}(\boldsymbol{x})$ is smaller than $1/2$ with high probability for labels $j$ whose conditional probability $\eta_j(\boldsymbol{x})$ is smaller than $t$. These labels can be handled easily in a regret analysis, since their estimates will be smaller than $1/2$. This implies that $\phi_{\ell_H}^{n,k}(\boldsymbol{x}) = \phi_{\ell_H}^*(\boldsymbol{x})$, so that the $k$-NN classifier does not suffer any regret in this case. By virtue of the sparsity assumption, there are at least $m - s$ such labels. We will present this claim in Lemma B.2.

- As a next step, we upper bound the probability of observing an instance such that the regression function has high error on labels from $T^s(\eta(\boldsymbol{x}))$, that is, when the error bound of Lemma B.1 fails, and the regression function estimates $\hat{\eta}_j^{n,k}(\boldsymbol{x})$ for $j \notin T^s(\eta(\boldsymbol{x}))$ are higher than $1/2$, that is when Lemma B.2 fails. We cannot upper bound the regret for these instances by other than a trivial upper bound such as $1$. So a small upper bound for the probability of these instances, which will be $\delta$, implies that the learner suffers only at most $\delta$ regret overall on such instances. This will show up as an additive term in the regret bound. We present this result in Lemma B.4.

- As a last step, we carry out a standard regret analysis focusing on $T^s(\eta(\boldsymbol{x}))$, since we know that labels $j \notin T^s(\eta(\boldsymbol{x}))$ add at most $\delta$ to the regret. Using the margin condition given in Assumption 4.2, we can have fast rate $1/n$ like in the analysis of plug-in classifiers of [2]. In addition, the regret bound scales with $s/m$, since we eliminated the rest of the labels, and it is enough to focus only on those from $T^s(\eta(\boldsymbol{x}))$.

We start by bounding the error of the $k$-NN regression function on labels from $T^s(\boldsymbol{x})$.

**Lemma B.1.** *Suppose that Assumption 5.1 holds. Given $\boldsymbol{x} \in \mathcal{X}, k \in [n]$ with $k \geq 4\log(2/\delta)$, with probability at least $1 - \delta$ over $\mathcal{D}$, we have*

$$\max_{j \in T^s(\eta(\boldsymbol{x}))} \left\{ \left| \hat{\eta}_j^{n,k}(\boldsymbol{x}) - \eta_j(\boldsymbol{x}) \right| \right\} \leq C_\lambda \cdot \left( \frac{2k}{n} \right)^\lambda + \sqrt{\frac{\log(4s/\delta)}{2k}} \ .$$

*Proof.* Let us introduce the notation $\mathbf{P}'( \ . \ ) = \mathbf{P}\left( \ . \ | \ \{\boldsymbol{X}_i\}_{i \in [n]} \right)$. First we condition on $\{\boldsymbol{X}_i\}_{i \in [n]}$ and apply Hoeffding's lemma to obtain

$$\mathbf{P}'\left( \max_{j \in T^s(\eta(\boldsymbol{x}))} \left\{ \left| \hat{\eta}_j^{n,k}(\boldsymbol{x}) - \frac{1}{k} \sum_{q \in [k]} \eta^j(\boldsymbol{X}^{\tau_{n,q}(\boldsymbol{x})}) \right| \right\} > \xi \right)$$

$$\leq \sum_{j \in T^s(\eta(\boldsymbol{x}))} \mathbf{P}'\left( \left| \frac{1}{k} \sum_{q \in [k]} \left( Y_j^{\tau^{n,q}(\boldsymbol{x})} - \eta_j(\boldsymbol{X}^{\tau_{n,q}(\boldsymbol{x})}) \right) \right| > \xi \right)$$

$$\leq 2s \cdot e^{-2k\xi^2} \ .$$

By taking $\xi = \sqrt{\log(4s/\delta)/2k}$ and marginalising over $\{\boldsymbol{X}_i\}_{i \in [n]}$ we see that the following bound holds with probability at least $1 - \delta/2$,

$$\max_{j \in T^s(\eta(\boldsymbol{x}))} \left\{ \left| \hat{\eta}_j^{n,k}(\boldsymbol{x}) - \frac{1}{k} \sum_{q \in [k]} \eta_j\left( \boldsymbol{X}^{\tau_{n,q}(\boldsymbol{x})} \right) \right| \right\} \leq \sqrt{\frac{\log(4s/\delta)}{2k}} \ .$$

To conclude the proof of the lemma we apply Corollary A.2 to infer that with probability at least $1 - \delta/2$ we have

$$\max_{j \in T^s(\eta(\boldsymbol{x}))} \left\{ \left| \eta_j(\boldsymbol{x}) - \frac{1}{k} \sum_{q \in [k]} \eta_j(\boldsymbol{X}^{\tau_{n,q}(\boldsymbol{x})}) \right| \right\} \leq C_\lambda \cdot \left( \frac{2k}{n} \right)^\lambda \ .$$

The lemma follows by applying a union bound and the triangle inequality. $\square$

As a next step, we bound the maximum of the $k$-NN regression function for labels $j \notin T^s(\eta(\boldsymbol{x}))$.

**Lemma B.2.** *Suppose that Assumption 5.1 and 4.1 hold. Given $\boldsymbol{x} \in \mathcal{X}$, and $k \in \mathbb{N}$ such that*

$$8\log(2m/\delta) \leq k \leq n/2 \ ,$$

*with probability at least $1 - \delta$ over $\mathcal{D}$ we have*

$$\max_{j \notin T^s(\eta(\boldsymbol{x}))} \left\{ \hat{\eta}_j^{n,k}(\boldsymbol{x}) \right\} \leq \max_{j \notin T^s(\eta(\boldsymbol{x}))} \left\{ \eta_j(\boldsymbol{x}) \right\} + C_\lambda \cdot \left( \frac{2k}{n} \right)^\lambda + \sqrt{\frac{8\log(2m/\delta)}{k}} \ .$$

*Moreover, under Assumption 4.1, with probability at least $1 - \delta$ over $\mathcal{D}$, we have*

$$\max_{j \notin T^s(\eta(\boldsymbol{x}))} \left\{ \hat{\eta}_j^{n,k}(\boldsymbol{x}) \right\} \leq t + C_\lambda \cdot \left( \frac{2k}{n} \right)^\lambda + \sqrt{\frac{8\log(2m/\delta)}{k}} \ .$$

*Proof.* Applying Corollary A.2, with probability at least $1 - \delta/2$, the following holds for all $q \in [k]$,

$$\max_{j \notin T^s(\eta(\boldsymbol{x}))} \left\{ \frac{1}{k} \sum_{q \in [k]} \eta_j(\boldsymbol{X}_{\tau_{n,q}(\boldsymbol{x})}) \right\} \leq \max_{j \notin T^s(\eta(\boldsymbol{x}))} \left\{ \eta_j(\boldsymbol{x}) + \left| \frac{1}{k} \sum_{q \in [k]} \eta_j(\boldsymbol{X}_{\tau_{n,q}(\boldsymbol{x})}) - \eta_j(\boldsymbol{x}) \right| \right\}$$

$$\leq \max_{j \notin T^s(\eta(\boldsymbol{x}))} \left\{ \eta_j(\boldsymbol{x}) \right\} + \max_{j \in [m]} \left\{ \left| \frac{1}{k} \sum_{q \in [k]} \eta_j(\boldsymbol{X}_{\tau_{n,q}(\boldsymbol{x})}) - \eta_j(\boldsymbol{x}) \right| \right\}$$

$$\leq \max_{j \notin T^s(\eta(\boldsymbol{x}))} \left\{ \eta_j(\boldsymbol{x}) \right\} + \frac{1}{k} \sum_{q \in [k]} \max_{j \in [m]} \left\{ \left| \eta_j(\boldsymbol{X}_{\tau_{n,q}(\boldsymbol{x})}) - \eta_j(\boldsymbol{x}) \right| \right\}$$

$$\leq \max_{j \notin T^s(\eta(\boldsymbol{x}))} \left\{ \eta_j(\boldsymbol{x}) \right\} + C_\lambda \cdot \left( \frac{2k}{n} \right)^\lambda \ .$$

Let us recall the notation $\mathbf{P}'(\ .\ )$ given in the proof of Lemma B.1. By conditioning on $\{\boldsymbol{X}_i\}_{i\in[n]}$ we have

$$\mathbf{P}'\left(\max_{j\notin T^s(\eta(\boldsymbol{x}))}\hat{\eta}_j(\boldsymbol{x}) - \max_{j\notin T^s(\eta(\boldsymbol{x}))}\frac{1}{k}\sum_{q\in[k]}\eta_j(\boldsymbol{X}^{\tau_{n,q}(\boldsymbol{x})}) > \xi\right) \tag{9}$$

$$\leq \mathbf{P}'\left(\max_{j\notin T^s(\eta(\boldsymbol{x}))}\left|\hat{\eta}_j(\boldsymbol{x}) - \frac{1}{k}\sum_{q\in[k]}\eta_j(\boldsymbol{X}^{\tau_{n,q}(\boldsymbol{x})})\right| > \xi\right) \tag{10}$$

$$= \mathbf{P}'\left(\max_{j\notin T^s(\eta(\boldsymbol{x}))}\left|\frac{1}{k}\sum_{q\in[k]}\left(Y_j^{\tau_{n,q}(\boldsymbol{x})} - \eta_j(\boldsymbol{X}^{\tau_{n,q}(\boldsymbol{x})})\right)\right| > \xi\right)$$

$$= \sum_{j\notin T^s(\eta(\boldsymbol{x}))}\mathbf{P}'\left(\left|\frac{1}{k}\sum_{q\in[k]}\left(Y_j^{\tau_{n,q}(\boldsymbol{x})} - \eta_j(\boldsymbol{X}^{\tau_{n,q}(\boldsymbol{x})})\right)\right| > \xi\right)$$

$$\leq 2(m-s)e^{-\frac{k\xi^2}{2(1+\xi/3)}} \tag{11}$$

$$\leq 2me^{-\frac{k\xi^2}{8}},$$

where (10) follows from the fact that $\|\boldsymbol{y}\|_\infty - \|\boldsymbol{y}'\|_\infty \leq \|\boldsymbol{y} - \boldsymbol{y}'\|_\infty$ for any vectors $\boldsymbol{y}, \boldsymbol{y}'$ from a normed space, and (11) follows from the Union bound and the Bernstein inequality.

By Assumption 4.1 we have $\max_{j\notin T^s(\eta(\boldsymbol{x}))}\eta_j(\boldsymbol{x}) \leq t$ which implies the second claim of lemma. $\qquad\square$

Lemma B.2 implies that if we set $k$ to be large enough, then all estimates for labels in $T^s(\eta(\boldsymbol{x}))$ will be smaller than $1/2$, thus the $k$-NN learner does not suffer any regret on these labels, and we can handle this case easily in the regret analysis. We summarize this observation in the following corollary.

**Corollary B.3.** *Suppose that Assumption 5.1 and 4.1 hold. Given $\boldsymbol{x} \in \mathcal{X}$ and $k \in \mathbb{N}$ such that*

$$16(1-2t)^{-2}\log(2m/\delta) \leq k \leq ((1-2t)/(8C_\lambda))^{1/\lambda}(n/2),$$

*with probability at least $1-\delta$ over $\mathcal{D}$, we have $\max_{j\notin T^s(\eta(\boldsymbol{x}))}\{\hat{\eta}_j^{n,k}(\boldsymbol{x})\} < 1/2$.*

As a next step, for a given sample $\mathcal{D}$ and $\delta \in (0,1)$ define

$$\mathcal{G}_\delta(\mathcal{D}) := \left\{\boldsymbol{x} \in \mathcal{X}\ :\ \forall j \in T^s(\eta(\boldsymbol{x})),\ \left|\hat{\eta}_j^{n,k}(\boldsymbol{x}) - \eta_j(\boldsymbol{x})\right| \leq C_\lambda \cdot \left(\frac{2k}{n}\right)^\lambda + \sqrt{\frac{\log(8s/\delta^2)}{2k}}\right\}$$

$$\cap \left\{\boldsymbol{x} \in \mathcal{X}\ :\ \forall j \notin T^s(\eta(\boldsymbol{x})),\ \hat{\eta}_j^{n,k}(\boldsymbol{x}) < \frac{1}{2}\right\}.$$

The set $\mathcal{G}_\delta(\mathcal{D}) \subseteq \mathcal{X}$ contains those instances in the feature space for which we can compute an upper bound on the regret with a non-trivial term, since the error of the $k$-NN regression estimate is small for the top-$s$ $T^s(\eta(\boldsymbol{x}))$. In addition, the $k$-NN regression estimates are smaller than $1/2$ for $j \notin T^s(\eta(\boldsymbol{x}))$. So we would like to upper bound the probability of observing an instances from $\mathcal{X}\backslash\mathcal{G}_\delta(\mathcal{D})$, because for these instance, we can have only trivial upper bound on the regret of the $k$-NN learner. Notice that $\mu\left(\mathcal{X}\backslash\mathcal{G}_\delta(\mathcal{D})\right)$ is a random variable which depends on the selection of random sample $\mathcal{D}$. The next lemma provides a high-probability upper bound on the marginal probability measure $\mu$ of $\mathcal{X}\backslash\mathcal{G}_\delta(\mathcal{D})$.

**Lemma B.4.** *Suppose that Assumption 5.1 and 4.1 hold. Moreover suppose that*

$$32(1-2t)^{-2}\log(2/\delta) \leq k \leq ((1-2t)/(8C_\lambda))^{1/\lambda}(n/2).$$

*Then with probability at least $1-\delta$ over $\mathcal{D}$ we have $\mu\left(\mathcal{X}\backslash\mathcal{G}_\delta(\mathcal{D})\right) \leq \delta$.*

*Proof.* By Lemma B.1 combined with Corollary B.3 and a union bound, we see that for each $\boldsymbol{x} \in \mathcal{X}$, $\mathbf{P}\left(\boldsymbol{X} \notin \mathcal{G}_\delta(\mathcal{D})\right) \leq \delta^2$. Hence, by Fubini's theorem we have

$$\mathbf{E}\left(\mu\left(\mathcal{X}\backslash\mathcal{G}_\delta(\mathcal{D})\right)\right) = \mathbf{E}\left(\int \mathbb{I}\{\boldsymbol{x} \notin \mathcal{G}_\delta(\mathcal{D})\}d\mu(\boldsymbol{x})\right) = \int \mathbf{E}\left(\mathbb{I}\{\boldsymbol{x} \notin \mathcal{G}_\delta(\mathcal{D})\}\right)d\mu(\boldsymbol{x}) \leq \delta^2.$$

The conclusion follows by Markov's inequality. $\qquad\square$

Now we present the proof of Theorem 5.2.

*Proof of Theorem 5.2.* By Lemma B.4 it suffices to take

$$\Delta(n, k, s, \delta) = C_\lambda \cdot (2k/n)^\lambda + \sqrt{\log(8s/\delta)/k}$$

and show that $\mu\left(\mathcal{X}\backslash\mathcal{G}_\delta(\mathcal{D})\right) \leq \delta$ implies

$$\text{Reg}_{\ell_H}(\phi_{\ell_H}^{n,k}) \leq \frac{2s}{m} \cdot C_\alpha \cdot \Delta(n, k, s, \delta)^{1+\alpha} + \delta \, .$$

Take $\boldsymbol{x} \in \mathcal{G}_\delta(\mathcal{D})$ and recall that $(\phi_{\ell_H}^*(\boldsymbol{x}))_j = \mathbb{I}\{\eta_j(\boldsymbol{x}) \geq 1/2\}$ and $(\phi_{\ell_H}^{n,k}(\boldsymbol{x}))_j = \mathbb{I}\{\hat{\eta}_j^{n,k}(\boldsymbol{x}) \geq 1/2\}$. We consider the two cases $j \in T^s(\eta(\boldsymbol{x}))$ and $j \notin T^s(\eta(\boldsymbol{x}))$.

Suppose $j \in T^s(\eta(\boldsymbol{x}))$. By construction, since $\boldsymbol{x} \in \mathcal{G}_\delta(\mathcal{D})$ and $(\phi_{\ell_H}^{n,k}(\boldsymbol{x}))_j \neq (\phi_{\ell_H}^*(\boldsymbol{x}))_j$, we must have

$$|\eta_j(\boldsymbol{x}) - 1/2| \leq C_\lambda \cdot (2k/n)^\lambda + \sqrt{\log(8s/\delta)/k} \, .$$

Hence, we have

$$\mathbb{I}\{(\phi_{\ell_H}^{n,k}(\boldsymbol{x}))_j \neq (\phi_{\ell_H}^*(\boldsymbol{x}))_j\} \cdot |2\eta_j(\boldsymbol{x}) - 1| \leq 2 \cdot \mathbb{I}\{(\phi_{\ell_H}^{n,k}(\boldsymbol{x}))_j \neq (\phi_{\ell_H}^*(\boldsymbol{x}))_j\} \cdot \left|\eta_j(\boldsymbol{x}) - \frac{1}{2}\right|$$

$$\leq 2 \cdot \mathbb{I}\{|\eta_j(\boldsymbol{x}) - 1/2| \leq \Delta(n, k, s, \delta)\} \cdot \Delta(n, k, s, \delta) \, .$$

On the other hand, when $j \notin T^s(\eta(\boldsymbol{x}))$ we have $\eta_j(\boldsymbol{x}) \leq t < 1/2$ so that $(\phi_{\ell_H}^*(\boldsymbol{x}))_j = 0$. Since $\boldsymbol{x} \in \mathcal{G}_\delta(\mathcal{D})$ we have $\hat{\eta}_j^{n,k}(\boldsymbol{x}) < 1/2$ so $(\phi_{\ell_H}^{n,k}(\boldsymbol{x}))_j = (\phi_{\ell_H}^*(\boldsymbol{x}))_j$. Thus, for $j \notin T^s(\eta(\boldsymbol{x}))$ we have $\mathbb{I}\{(\phi_{\ell_H}^{n,k}(\boldsymbol{x}))_j \neq (\phi_{\ell_H}^*(\boldsymbol{x}))_j\} \cdot |2\eta_j(\boldsymbol{x}) - 1| = 0$. As a consequence, for $\boldsymbol{x} \in \mathcal{G}_\delta(\mathcal{D})$ we can write

$$\frac{1}{m} \sum_{j \in [m]} \mathbb{I}\{(\phi_{\ell_H}^{n,k}(\boldsymbol{x}))_j \neq (\phi_{\ell_H}^*(\boldsymbol{x}))_j\} \cdot |2\eta_j(\boldsymbol{x}) - 1|$$

$$\leq \frac{2}{m} \sum_{j \in T^s(\eta(\boldsymbol{x}))} \mathbb{I}\{|\eta_j(\boldsymbol{x}) - 1/2| \leq \Delta(n, k, s, \delta)\} \times \Delta(n, k, s, \delta)$$

$$\leq \frac{2s}{m} \cdot \mathbb{I}\left\{\boldsymbol{x} \in \mathcal{X} \ : \ \min_{j \in [m]}\{|\eta_j(\boldsymbol{x}) - 1/2|\} \leq \Delta(n, k, s, \delta)\right\} \times \Delta(n, k, s, \delta) \, .$$

Further, by Assumption 4.2 we have

$$\text{Reg}_{\ell_H}(\phi_{\ell_H}^{n,k}) \leq \frac{2s}{m} \cdot \int_{\mathcal{X}} \mathbb{I}\left\{\min_{j \in [m]}\{|\eta_j(\boldsymbol{x}) - 1/2|\} \leq \Delta(n, k, s, \delta)\right\} \times \Delta(n, k, s, \delta) d\mu(\boldsymbol{x})$$

$$\leq \frac{2s}{m} \cdot C_\alpha \cdot \Delta(n, k, s, \delta)^{1+\alpha} + \mu(\mathcal{X}\backslash\mathcal{G}_\delta(\mathcal{D}))$$

$$\leq \frac{2s}{m} \cdot C_\alpha \cdot \Delta(n, k, s, \delta)^{1+\alpha} + \delta \, .$$

This completes the proof of the theorem. $\qquad\square$

## C Proof of Theorem 5.7

The proof of Theorem 5.7 has three main steps:

- First, similar to Lemma B.1, we upper bound the error of the $k$-NN regression function for all labels. This result is presented in Corollary C.1.

- Similar to the proof of Theorem 5.2, we upper bound by $\delta$ the probability of observing an instance for which the error of the $k$-NN regression function is large, as for those instances we can only have trivial upper bounds.

- Finally, we note that if the error of the $k$-NN regression function for $\hat{\eta}_{(\kappa)}(\boldsymbol{x})$ and $\hat{\eta}_{(\kappa+1)}(\boldsymbol{x})$ is upper bounded by a term $\Delta$, then the classifier suffers at most $2\Delta$ $\ell_{@\kappa}$-regret on $\boldsymbol{x}$. Using the margin condition, we can upper bound the probability of instances for which $\eta_\kappa(\boldsymbol{x}) - \eta_{\kappa+1}(\boldsymbol{x}) \leq \Delta$ by $K_\beta \Delta^{1+\beta}$, which results in the regret bound of Theorem 5.7.

Corollary C.1 below is a consequence of Lemma B.1 which gives an error of the $k$-NN regression function for every label.

**Corollary C.1.** *Suppose that Assumption 5.1 holds. Given $x \in \mathcal{X}, n/2 \geq k \in [n]$ with $k \geq 4\log(2/\delta)$, the following holds with probability at least $1 - \delta$ over $\mathcal{D}$:*

$$\max_{j \in [m]} \left\{ \left| \hat{\eta}_j^{n,k}(\boldsymbol{x}) - \eta_j(\boldsymbol{x}) \right| \right\} \leq C_\lambda \cdot \left( \frac{2k}{n} \right)^\lambda + \sqrt{\frac{\log(4m/\delta)}{2k}} \ .$$

*Proof.* The proof is straightforward based on the proof of Lemma B.1. $\square$

With this upper bound on the pointwise error, we can prove Theorem 5.7.

*Proof of Theorem 5.7.* Let us define

$$\mathcal{G}_\delta = \mathcal{G}_\delta(\mathcal{D}) := \left\{ x \in \mathcal{X} : \max_{i \in [m]} |\eta_i(\boldsymbol{x}) - \hat{\eta}_i^{n,k}(\boldsymbol{x})| \leq \Delta(n,k,\delta) \right\} \ ,$$

where $\Delta(n,k,\delta) = \sqrt{\frac{\log(2m/\delta)}{k}} + \omega \left( \frac{2k}{n} \right)^k$. Note that $\mathcal{G}_\delta$ depends on $\mathcal{D}$ through the estimator $\hat{\eta}^{n,k}(\boldsymbol{x})$ which clearly depends on $\mathcal{D}$. Based on Corollary C.1, we have

$$\max_{j \in [m]} |\hat{\eta}_j^{n,k}(\boldsymbol{x}) - \eta_j(\boldsymbol{x})| \leq \Delta(n,k,\delta) \tag{12}$$

with probability $1 - \delta^2/3$ for any $\boldsymbol{x} \in \mathcal{X}$. Thus $\mathbb{E}[\mu(\mathcal{X} \setminus \mathcal{G}_\delta)] \leq \delta/3$, and based on Markov's inequality, it holds that $\mu(\mathcal{X} \setminus \mathcal{G}_\delta) \leq \delta$ with probability at least $1 - \delta/3$. Next let us rewrite the regret as

$$\text{Reg}_{\ell_{@\kappa}}(\phi_{\ell_{@\kappa}}^{n,k}) = \frac{1}{\kappa} \int_{\boldsymbol{x} \in \mathcal{X}} \left[ \sum_{i \in T^\kappa(\eta(\boldsymbol{x}))} \eta_i(\boldsymbol{x}) - \sum_{j \in T^\kappa(\hat{\eta}^{n,k}(\boldsymbol{x}))} \eta_j(\boldsymbol{x}) \right] d\mu(\boldsymbol{x})$$

$$= \frac{1}{\kappa} \int_{\mathcal{G}_\delta} \left[ \sum_{i \in T^\kappa(\eta(\boldsymbol{x}))} \eta_i(\boldsymbol{x}) - \sum_{j \in T^\kappa(\hat{\eta}^{n,k}(\boldsymbol{x}))} \eta_j(\boldsymbol{x}) \right] d\mu(\boldsymbol{x})$$

$$+ \frac{1}{\kappa} \int_{\mathcal{X} \setminus \mathcal{G}_\delta} \left[ \sum_{i \in T^\kappa(\eta(\boldsymbol{x}))} \eta_i(\boldsymbol{x}) - \sum_{j \in T^\kappa(\hat{\eta}^{n,k}(\boldsymbol{x}))} \eta_j(\boldsymbol{x}) \right] d\mu(\boldsymbol{x})$$

$$\leq \frac{1}{\kappa} \int_{\mathcal{G}_\delta} \left[ \sum_{i \in T^\kappa(\eta(\boldsymbol{x}))} \eta_i(\boldsymbol{x}) - \sum_{j \in T^\kappa(\hat{\eta}^{n,k}(\boldsymbol{x}))} \eta_j(\boldsymbol{x}) \right] d\mu(\boldsymbol{x}) + 2\mu(\mathcal{X} \setminus \mathcal{G}_\delta)$$

$$\leq \frac{1}{\kappa} \int_{\mathcal{G}_\delta} \left[ \sum_{i \in T^\kappa(\eta(\boldsymbol{x}))} \eta_i(\boldsymbol{x}) - \sum_{j \in T^\kappa(\hat{\eta}^{n,k}(\boldsymbol{x}))} \eta_j(\boldsymbol{x}) \right] d\mu(\boldsymbol{x}) + 2\delta$$

$$\leq \int_{\mathcal{G}_\delta} 2\Delta(n,k,\delta)d\mu(\boldsymbol{x}) + 2\delta \tag{13}$$

$$\leq \int_{\mathcal{X}} \mathbb{I}\{\eta_{(\kappa)}(\boldsymbol{x}) - \eta_{(\kappa+1)}(\boldsymbol{x}) < 2\Delta(n,k,\delta)\} \times 2\Delta(n,k,\delta)d\mu(\boldsymbol{x}) + 2\delta$$

$$\leq K_\beta \cdot (2\Delta(n,k,\delta))^{1+\beta} + 2\delta \ , \tag{14}$$

where (13) follows from the fact that if $\eta_{(\kappa)}(\boldsymbol{x}) - \eta_{(\kappa+1)}(\boldsymbol{x}) \geq 2\Delta(n,k,\delta)$ for any $\boldsymbol{x} \in \mathcal{G}_\delta$ then this implies that the conditional regret $\text{Reg}_{\ell_{@\kappa}}(\phi_{\ell_{@\kappa}}^{n,k} \,|\, \boldsymbol{x})$ is zero because of (12), while if $\eta_{(\kappa)}(\boldsymbol{x}) - \eta_{(\kappa+1)}(\boldsymbol{x}) < 2\Delta(n,k,\delta)$, then $\text{Reg}_{\ell_{@\kappa}}(\phi_{\ell_{@\kappa}}^{n,k} \,|\, \boldsymbol{x}) \leq 2\Delta(n,k,\delta)$. All these claims hold with probability at least $1 - \delta/2$. In the last step (14) we used Assumption 5.6. An application of the union bound concludes the proof. $\square$

# D  Proofs of Theorem 5.10 and Theorem 5.11

*Proof of Theorem 5.10.* For Hamming loss, given a binary problem $P$, let $g_p(P) := \tilde{P}$ denote the multi-label problem with

$$\eta_j(\boldsymbol{x}) = \begin{cases} \eta(\boldsymbol{x}), & \text{for } j = 1, \dots, s \\ 0, & \text{for } j = s + 1, \dots, m, \end{cases}$$

and, given sample $\mathcal{D} = \{(\boldsymbol{X}^i, Y^i)\}_{i \in [n]}$ from $P$, let $g_d(\mathcal{D}) = \{(\boldsymbol{X}^i, \tilde{\boldsymbol{Y}}^i)\}_{i \in [n]}$, where $\tilde{\boldsymbol{Y}}^i \in \{0, 1\}^m$ is the vector whose first $s$ components are set to $Y^i$, and the remaining $m - s$ set to 0. An (unbiased) sample from $P$ are mapped to (unbiased) sample from $g_p(P)$, and label independence can be enforced at some cost. In addition, for a classifier $\tilde{\phi} \in \mathcal{M}(\mathcal{X}, \mathcal{Y})$ we define $f(\tilde{\phi}) \in \mathcal{M}(\mathcal{X}, \{0, 1\})$ as

$$f(\tilde{\phi})(\boldsymbol{x}) = \mathbb{I}\{\left| [s] \cap \tilde{\phi}(\boldsymbol{x}) \right| \geq s/2\}$$

for $\boldsymbol{x} \in \mathcal{X}$. This immediately implies

$$\mathrm{Reg}_{\ell_H}^{\tilde{P}}\left(\tilde{\phi} \mid \boldsymbol{x}\right) \geq (\mathfrak{s}/2)\mathrm{Reg}_{\ell_{0-1}}^{P}\left(f(\tilde{\phi}) \mid \boldsymbol{x}\right) ,$$

as claimed. As a last step, it is easy to check that if we consider binary problem having binary margin condition

$$\mathbf{P}\left(|\eta(\boldsymbol{X}) - 1/2| < \xi\right) \leq C_\alpha \xi^\alpha$$

then this implies multi-label margin condition as defined in Assumption 4.2 with the same parameters. Any lower bound on expected regret applies to high probability lower bounds. In particular, the lower bound result contained in [2] (Theorem 3.5 therein) for expected regret can be stated in our setup as $Cn^{-\frac{\lambda(1+\alpha)}{2\lambda+1}}$, for some constant $C > 0$. This concludes the proof. □

*Proof of Theorem 5.11.* For Precision@$\kappa$, the lower bound reduction is very similar to the one we just presented for Hamming loss. In particular, given a binary problem $P$, let $g_p(P) := \tilde{P}$ denote the multi-label problem with

$$\eta_j(\boldsymbol{x}) = \begin{cases} \eta(\boldsymbol{x}), & \text{for } j = 1, \dots, \kappa \\ 1 - \eta(\boldsymbol{x}), & \text{for } j = \kappa + 1, \dots, 2\kappa \\ 0, & \text{for } j = 2\kappa + 1, \dots, m . \end{cases}$$

Correspondingly, given sample $\mathcal{D} = \{(\boldsymbol{X}^i, Y^i)\}_{i \in [n]}$ from $P$, let $g_d(\mathcal{D}) = \{(\boldsymbol{X}^i, \tilde{\boldsymbol{Y}}^i)\}_{i \in [n]}$, where $\tilde{\boldsymbol{Y}}^i \in \{0, 1\}^m$ is the vector whose first $\kappa$ components are set to $Y^i$, the next $\kappa$ components (indices from $\kappa + 1$ to $2\kappa$) set to $(1 - Y^i)$, and the remaining $m - 2\kappa$ components are set to 0. Note that this maps (unbiased) samples from $P$ to (unbiased) samples from $g_p(P)$. (In case we want, we can enforce label independence at the cost of ending up with $1/(2\kappa)$ of the sample size for $g_p(P)$.) Finally, for a classifier $\tilde{\phi} \in \mathcal{M}(\mathcal{X}, \mathcal{Y})$ define

$$f(\tilde{\phi}) \in \mathcal{M}(\mathcal{X}, \{0, 1\})$$

as

$$f(\tilde{\phi})(\boldsymbol{x}) = \mathbb{I}\{\left| [\kappa] \cap \tilde{\phi}(\boldsymbol{x}) \right| \geq \kappa/2\}$$

for $\boldsymbol{x} \in \mathcal{X}$.

Now, the optimal classifier for $P$ is $\phi(\boldsymbol{x}) = \mathbb{I}\{\eta(\boldsymbol{x}) > 1/2\}$. Also note that the (conditional) regrets are related as follows:

$$\mathrm{Reg}_{\ell_{@\kappa}}^{\tilde{P}}\left(\tilde{\phi} \mid \boldsymbol{x}\right) \geq \begin{cases} \frac{2}{\kappa}\left(\eta(\boldsymbol{x}) - \frac{1}{2}\right)\left|\tilde{\phi}(\boldsymbol{x}) \setminus [\kappa]\right| & \text{if } \eta(\boldsymbol{x}) > 1/2 \\ \frac{2}{\kappa}\left(\frac{1}{2} - \eta(\boldsymbol{x})\right)\left|[\kappa] \cap \tilde{\phi}(\boldsymbol{x})\right| & \text{if } \eta(\boldsymbol{x}) \leq 1/2 \end{cases} \geq \frac{1}{2}\mathrm{Reg}_{\ell_{0-1}}^{P}\left(f(\tilde{\phi}) \mid \boldsymbol{x}\right).$$

Finally note that $f$ is a surjective mapping, and that $\tilde{\phi}$ is optimal for $g_p(P)$ iff $f(\tilde{\phi})$ is optimal for $P$ and $\tilde{\phi}(\boldsymbol{x}) \in \{[\kappa], [2\kappa] \setminus [\kappa]\}$ for all $\boldsymbol{x} \in \mathcal{X}$.

Consequently, any multi-label learner $A$ can be used as a binary learner with worst case regret upper bounded by half the worst case regret of $A$. Therefore, regret lower bounds for the binary problem automatically infer $(1/2)$ times the same lower bound for the multi-label learning problem. As a last step, it is easy to check that if we consider binary problem having binary margin condition

$$\mathbf{P}\left(|\eta(\boldsymbol{X}) - 1/2| < \xi\right) \leq C_\alpha \xi^\alpha$$

then this implies Precision@$\kappa$ margin condition

$$\mathbf{P}\left(\eta_{(\kappa)}(\boldsymbol{X}) - \eta_{(\kappa+1)}(\boldsymbol{X}) < 2\xi\right) \leq K_\beta (2\xi)^\beta$$

with $C_\alpha = K_\beta$ and $\beta = \alpha$. Using the same argument as for Hamming loss, any lower bound on expected regret implies high probability lower bounds. In particular, the lower bound result contained in [2] (Theorem 3.5 therein) for expected regret can be stated in our setup as $Cn^{-\frac{\lambda(1+\alpha)}{2\lambda+1}}$, for some constant $C > 0$, thereby concluding the proof. $\qquad\square$

# E   Multi-label extension of [3]

As a preliminary step, we start by presenting a MLC extension of Theorem 1 in [3], along with an improvement under low noise conditions.

Recall that in [3], it is shown that for each classification-calibrated loss, there exists a nondecreasing function $\psi : [0,1] \mapsto [0,\infty)$ that provides an upper bound on the regret of the 0-1 loss in terms of the regret of the surrogate loss, i.e., $\psi(\text{Reg}_{0\text{-}1}(f)) \leq \text{Reg}_\ell(f)$ for all measurable function $f : \mathcal{X} \mapsto \mathbb{R}$. Here, we extend their 0-1 loss result to Hamming loss in MLC setting.

First, we provide relevant definitions from [3]. Let $\mathcal{C}_\eta(z)$ be a *generic conditional $\ell$-risk*, which is defined as

$$\mathcal{C}_\eta(z) = \eta\ell(z) + (1-\eta)\ell(-z)\,.$$

The optimal conditional $\ell$-risk is

$$H(\eta) = \inf_{z\in\mathbb{R}} \mathcal{C}_\eta(z)\,.$$

The optimal conditional $\ell$-risk under the condition that the sign of $z$ disagrees with the sign of $2\eta - 1$ is,

$$H^-(\eta) = \inf_{z\in\mathbb{R}\,:\,z(2\eta-1)<0} \mathcal{C}_\eta(z)\,.$$

Then, given loss $\ell : \mathbb{R} \mapsto [0,\infty)$, the function $\psi : [-1,1] \mapsto [0,\infty)$ is defined by $\psi = \tilde{\psi}^{**}$, where

$$\tilde{\psi}(\theta) = H^-\left(\frac{1+\theta}{2}\right) - H\left(\frac{1+\theta}{2}\right)\,,$$

and $g^{**} : [-1,1] \mapsto R$ is the Fenchel–Legendre biconjugate of $g : [-1,1] \mapsto R$.

Next, we introduce new per label $j \in [m]$ notations for scoring functions $f : \mathcal{X} \mapsto \mathbb{R}^m$.

$$\text{Risk} \qquad R(\phi_f) = \frac{1}{m}\sum_{j=1}^m R_j(\phi_f) = \frac{1}{m}\sum_{j=1}^m \mathbf{E}\,\mathbb{I}\{\text{sgn}(f_j(\boldsymbol{X})) \neq \boldsymbol{Y}_j\}$$

$$\text{Bayes risk} \qquad R^* = \frac{1}{m}\sum_{j=1}^m R_j{}^* = \frac{1}{m}\sum_{j=1}^m \inf_f R_j(f)$$

$$\ell\text{-risk} \qquad R_\ell(f) = \frac{1}{m}\sum_{j=1}^m R_{\ell,j}(f_j) = \frac{1}{m}\sum_{j=1}^m \mathbf{E}\ell(\boldsymbol{Y}_j f_j(\boldsymbol{X}))$$

$$\text{Bayes } \ell\text{-risk} \qquad R_\ell^* = \frac{1}{m}\sum_{j=1}^m R_{\ell,j}^* = \frac{1}{m}\sum_{j=1}^m \inf_f R_{\ell,j}(f)$$

**Theorem E.1** (Hamming loss extension of Theorem 1 in [3])**.**

1. *For any nonnegative loss function $\ell$, any measurable function $f : \mathcal{X} \mapsto \mathbb{R}^m$, and probability distribution $P^j$, $j \in [m]$, on $\mathcal{X} \times \{\pm 1\}$,*

$$\psi(\text{Reg}_{\ell_H}(\phi_f)) \leq \text{Reg}_\ell(f).$$

2. *Suppose that $|\mathcal{X}| \geq 2$. For any nonnegative loss function $\ell$, any $\epsilon > 0$, and any $\theta \in [0,1]$, there is a probability distribution on $\mathcal{X} \times \{\pm 1\}$ and a function $f : \mathcal{X} \mapsto \mathbb{R}^m$ such that*

$$\text{Reg}_{\ell_H}(\phi_f) = \theta$$

   *and*

$$\psi(\theta) \leq \text{Reg}_\ell(f) \leq \psi(\theta) + \epsilon$$

3. *The following conditions are equivalent:*

   (a) *$\ell$ is classification-calibrated.*
   (b) *For every sequence of measurable functions $\{f^i\}$, such that $f^i : \mathcal{X} \mapsto \mathbb{R}^m$ and every probability distribution on $\mathcal{X} \times \{\pm 1\}^m$ with respect to which risk is computed,*

$$R_\ell(f^i) \to R_\ell^* \text{ implies that } R(\phi_{f^i}) \to R^*$$

*Proof.*      1. $\psi$ is a convex function from the definition. Thus,

$$\psi(\text{Reg}_{\ell_H}(\phi_f)) = \psi\left( \frac{1}{m} \sum_j R_j(\phi_{f_j}) - \frac{1}{m} \sum_j R_j{}^* \right)$$

$$= \psi\left( \frac{1}{m} \sum_j \left( R_j(\phi_{f_j}) - R_j^* \right) \right)$$

$$\leq \frac{1}{m} \sum_j \psi\left( R_j(\phi_{f_j}) - R_j^* \right) \tag{15}$$

$$\leq \frac{1}{m} \sum_j \left( R_{\ell,j}(f_j) - R_{\ell,j}^* \right) \tag{16}$$

$$= \text{Reg}_\ell(f) ,$$

where (15) is from Jensen's inequality and (16) is from Theorem 1.1 in [3].

2. This follows from Theorem 1 part 2 in [3]; $\psi(\theta) \leq \text{Reg}_\ell(f)$ follows from part 1 of the same theorem. To show

$$\text{Reg}_\ell(f) \leq \psi(\theta) + \epsilon , \tag{17}$$

we use the same construction as in the proof in Theorem 1.2 in [3] for each label $j \in [m]$ individually. Then we have

$$R_{\ell,j}(f_j) - R_{\ell,j}^* \leq \psi(\theta) + \epsilon . \tag{18}$$

Adding (18) for all $j \in [m]$ and dividing by $m$ gives us (17).

3. We first show that (a) $\to$ (b), and then that (b) $\to$ (a).

(a) $\to$ (b): Consider the case where $\ell$ is classification-calibrated. Then from Theorem 1.3.c in [3], for each $j \in [m]$, every sequence of measurable functions $f_j^i : \mathcal{X} \mapsto \mathbb{R}$ indexed by $i$, and every probability distribution $P_j$ on $\mathcal{X} \times \{\pm 1\}$,

$$R_{\ell,j}(f_j^i) \to R_{\ell,j}^* \text{ implies that } R_j(\phi_{f_j^i}) \to R_j^* . \tag{19}$$

Assume that

$$R_\ell(f^i) \to R_\ell{}^* \tag{20}$$

Since $R_\ell^*$ is the sum of the infima of $R_{\ell,j}$, (20) implies that

$$j \in [m], \ R_{\ell,j}(f_j^i) \to R_{\ell,j}^* .$$

Summing this up for all $j$ and using (19) gives us $R(\phi_{f^i}) \to R^*$.

(b)$\rightarrow$ (a): We follow the same construct as the proof of Theorem 1.3 in [3]. Consider the case where the statement of Theorem 1.3.b holds, and $R_\ell(f^i) \rightarrow R_\ell^*$. Assume $\ell$ is not classification calibrated. Then, for a fixed $\eta \neq \frac{1}{2}$, there exists a sequence $\{\alpha_i\}$ that both of the following conditions hold.

$$\alpha_i \cdot \mathrm{sgn}\left(\eta - \frac{1}{2}\right) = -1 \quad \forall i \tag{21}$$

$$C_\eta(\alpha_i) \rightarrow H(\eta)$$

For a fixed $x \in \mathcal{X}$, let all the probability distributions $P_j$ for labels $j \in [m]$ be such that $P(X = x) = 1$ and $P(Y_j = 1 \mid X = x) = \eta$. Let $f^i(x) = \alpha_i$. Then $\lim_{i \rightarrow \infty} R_j(f^i) > R_j^*$ from (21). This contradicts Theorem 1.3.b in that paper, in that $R(\phi_{f^i}) \rightarrow R^*$ should hold for any distributions and function $f^i$.

This concludes the proof. $\qquad\square$

## F  Proof of Theorem 6.2

Given the results in Appendix E, we are ready to prove Theorem 6.2.

Consider fixed $F : \mathcal{X} \mapsto \mathbb{R}^m$. Let $\mathcal{E}_j(x) = \mathbb{I}\{f_j(x)(\eta_j(x) - \frac{1}{2}) < 0]\}|2\eta_j(x) - 1|$ be the excessive risk for label $j$ and instance $x$. Note that $R(f) - R^* = \frac{1}{m}\sum_j \mathbf{E}[\mathcal{E}_j(X)]$. Let $j_i(x)$ be the label $j$ such that $\mathcal{E}_j(x)$ is the $i$th largest, e.g., $\mathcal{E}_{j_1(x)}(x) \geq \mathcal{E}_{j_2(x)}(x) \geq \cdots \geq \mathcal{E}_{j_m(x)}(x)$. Let $\tilde{R}_i(f) = \mathbf{E}[\mathcal{E}_{j_i(X)}(X)]$ be the excessive risk if the $i$-th the largest error label is chosen per instance.

Let $S_i = \{x | f_{j_i(x)}(x)\left(\eta_{j_i(x)} - \frac{1}{2}\right) < 0\}$, and notice that

1. $S_1 = \bigcup_j \{x \mid f_j(x)\left(\eta_j(x) - \frac{1}{2}\right) < 0\}$.
2. $S_i \supseteq S_j$ if $i \leq j$.

**Lemma F.1.** *If Assumption (4.2) holds, then for $\Gamma = (C_\alpha)^{1/(\alpha+1)}(1+\alpha)/(\alpha)^{\alpha/(1+\alpha)}$, for all $i \in [m]$, and all measurable $f : \mathcal{X} \mapsto \{\pm 1\}^m$,*
$$Pr\left(S_i\right) \leq \Gamma(\tilde{R}_i(f))^{\frac{\alpha}{1+\alpha}} .$$

*Proof.*

$$\tilde{R}_i(F) = \int_{S_i} \mathcal{E}_{j_i(X)}(X)dP(X)$$

$$\geq \int_{S_i} \mathbb{I}\{|2\eta_{j_i(X)}(X) - 1| > \epsilon\} \cdot \epsilon \, dP(X)$$

$$= \epsilon\left(\Pr(S_i) - \int_{S_i} \mathbb{I}\{0 < |2\eta_{j_i(X)}(X) - 1| \leq \epsilon\}dP(X)\right)$$

$$\geq \epsilon\left(\Pr(S_i) - \int \mathbb{I}\{\exists j, 0 < |2\eta_j(X) - 1| \leq \epsilon\}dP(X)\right)$$

$$\geq \epsilon\left(\Pr(S_i) - \Pr\left(\bigcup_j \left\{X \,\middle|\, 0 < |2\eta_j(X) - 1| \leq \epsilon\right\}\right)\right)$$

$$\geq \epsilon\left(\Pr(S_i) - C_\alpha\epsilon^\alpha\right) . \tag{22}$$

By setting $\epsilon = (\Pr(S_i)/(C_\alpha(1+\alpha)))^{1/\alpha}$, we get

$$(22) = \left(\frac{\Pr(S_i)}{C_\alpha(1+\alpha)}\right)^{1/\alpha}\left(\Pr(S_i) - \frac{\Pr(S_i)}{1+\alpha}\right) = \left(\frac{1}{(C_\alpha(1+\alpha))^{1/\alpha}}\right)\frac{\alpha}{1+\alpha}\Pr(S_i)^{(1+\alpha)/\alpha} .$$

Rearranging terms yields

$$\Pr(S_i) \leq (C_\alpha)^{1/(\alpha+1)}(1+\alpha)/(\alpha)^{\alpha/(1+\alpha)}\tilde{R}_i(F)^{\alpha/(1+\alpha)} , \tag{23}$$

as claimed. $\qquad\square$

Using this lemma, we can prove Theorem 6.2 as follows.

*Proof of Theorem 6.2.* For notational brevity, we will use $\gamma = \frac{\alpha}{\alpha+1}$. The regret is

$$R(f) - R^* \leq \frac{1}{m}\mathbf{E}\sum_j \mathbb{I}\{|2\eta_j(\boldsymbol{X}) - 1| < \epsilon\}\mathcal{E}_j + \frac{1}{m}\mathbf{E}\sum_j \mathbb{I}\{|2\eta_j(\boldsymbol{X}) - 1| \geq \epsilon\}\mathcal{E}_j \ .$$

For the first term in the RHS we can write

$$\frac{1}{m}\mathbf{E}\sum_j \mathbb{I}\{|2\eta_j(\boldsymbol{X}) - 1| < \epsilon\}\mathcal{E}_j$$

$$= \frac{1}{m}\mathbf{E}\sum_j \mathbb{I}\{|2\eta_j(\boldsymbol{X}) - 1| < \epsilon\}\mathbb{I}\{f_j(\boldsymbol{X})(\eta_j(\boldsymbol{X}) - 1/2) < 0\}|2\eta_j(\boldsymbol{X}) - 1|$$

$$\leq \frac{\epsilon}{m}\mathbf{E}\sum_j \mathbb{I}\{|2\eta_j(\boldsymbol{X}) - 1| < \epsilon\}\mathbb{I}\{f_j(\boldsymbol{X})(\eta_j(\boldsymbol{X}) - 1/2) < 0\} \ .$$

Let $\epsilon < 1 - 2t$. Then, per $\boldsymbol{X}$, there are at most only $s$ number of classes $j$ that satisfies $|2\eta_j(\boldsymbol{X}) - 1| < \epsilon$, and we can upper bound this term by using the largest $s$ number of $j$'s, which are $j_1(\boldsymbol{X}), j_2(\boldsymbol{X}), \ldots, j_s(\boldsymbol{X})$. Then

$$\frac{\epsilon}{m}\mathbf{E}\sum_j \mathbb{I}\{|2\eta_j(\boldsymbol{X}) - 1| < \epsilon\}\mathbb{I}\{f_j(\boldsymbol{X})(\eta_j(\boldsymbol{X}) - 1/2) < 0\}$$

$$\leq \frac{\epsilon}{m}\mathbf{E}\sum_{i=1}^s \mathbb{I}\{f_{j_i(\boldsymbol{X})}(\boldsymbol{X})(\eta_{j_i(\boldsymbol{X})} - 1/2) < 0\}$$

$$= \frac{\epsilon}{m}\sum_{i=1}^s \Pr(S_i)$$

$$\leq \frac{\Gamma\epsilon}{m}\sum_{i=1}^s (\tilde{R}_i(f))^\gamma$$

$$= \Gamma\epsilon\mathfrak{s}\sum_{i=1}^s \frac{1}{s}(\tilde{R}_i(f))^\gamma$$

$$\leq \Gamma\epsilon\mathfrak{s}\left(\frac{1}{s}\sum_{i=1}^s \tilde{R}_i(f)\right)^\gamma$$

$$\leq \Gamma\epsilon\mathfrak{s}\left(\frac{1}{s} \cdot m(\text{Reg}_{\ell_H}(\phi_f))\right)^\gamma \ .$$

As in proof of the Theorem 3 in [3], we upper bound the second term by

$$\frac{\epsilon}{\psi(\epsilon)}\left(R_\ell(f) - R_\ell{}^*\right),$$

and choose

$$\epsilon = \frac{1}{2\Gamma}(1/\mathfrak{s})^{1-\gamma}\left(R(f) - R^*\right)^{1-\gamma} \ .$$

Rearranging terms and observing that by (23) $\Gamma = C_\alpha^{1/(\alpha+1)}(\alpha + 1)/\alpha^{\alpha/(\alpha+1)}$ concludes the proof.

$\square$

# G    Proof of Theorem 6.7

*Proof.* Let $g_k(\boldsymbol{x})$ be the value of centered classifier $f$ for the label with the $k$th largest conditional probability, i.e., corresponding to $\eta_{[k]}(\boldsymbol{x})$. For notational convenience, we drop the dependence on $\boldsymbol{x}$ when clear from the context. We prove the second inequality, but the same argument applies to the first one. We can write

$$\text{Reg}_{\ell@\kappa}@(T^\kappa(f)) = \frac{1}{\kappa}\mathbf{E}\left[\sum_{k\in[\kappa]}\eta_{[k]}\mathbb{I}\{g_k \leq f_{[\kappa+1]}\} + \sum_{k\in[m]/[\kappa]}-\eta_{[k]}\mathbb{I}\{g_k > f_{[\kappa+1]}\}\right]$$

$$\leq \frac{1}{\kappa}\mathbf{E}\left[\sum_{k\in[\kappa]}\eta_{[k]}\mathbb{I}\{g_k < 0\} - \sum_{k\in[m]/[\kappa]}\eta_{[k]}\mathbb{I}\{g_k > 0\}\right] . \tag{24}$$

We now introduce pairing of labels. We pair each $k \in [\kappa]$, a label whose conditional probability is the $k$th largest, i.e., $\eta_{[k]}$, with a different label $k'' \in [m]/[\kappa]$ such that $g_{k''} > 0$ if and only if $g_k < 0$. Such pairs exist because $|\{k \in [\kappa] \,|\, g_k < 0\}| = |\{k \in [m]/[\kappa] \,|\, g_k > 0\}|$. Then

$$(24) = \frac{1}{\kappa}\mathbf{E}\left[\sum_{k\in[\kappa]}\left(\eta_{[k]}\mathbb{I}\{g_k < 0\} - \eta_{k''}\mathbb{I}\{g_{k''} > 0\}\right)\right] . \tag{25}$$

Now, for each $k$, if $\eta_{[k]} > \frac{1}{2}$, then

$$\frac{1}{2}\left(2\eta_{[k]}\mathbb{I}\{g_k < 0\} - 2\eta_{k''}\mathbb{I}\{g_{k''} > 0\}\right)$$

$$= \frac{1}{2}\left((2\eta_{[k]} - 1)\mathbb{I}\{g_k < 0\} + \mathbb{I}\{g_k < 0\} + (1 - 2\eta_{k''})\mathbb{I}\{g_{k''} > 0\}\right) - \mathbb{I}\{g_{k''} > 0\}\right)$$

$$= \frac{1}{2}\left((2\eta_{[k]} - 1)\mathbb{I}\{g_k < 0\} + (1 - 2\eta_{k''})\mathbb{I}\{g_{k''} > 0\}\right)$$

Otherwise,

$$\frac{1}{2}\left(2\eta_{[k]}\mathbb{I}\{g_k < 0\} - 2\eta_{k''}\mathbb{I}\{g_{k''} > 0\}\right)$$

$$= \frac{1}{2}\left((2\eta_{[k]} - 2\eta_{[k]})\mathbb{I}\{g_k < 0\} + 2\eta_{[k]}\mathbb{I}\{g_{k''} > 0\} - 2\eta_{k''}\mathbb{I}\{g_{k''} > 0\}\right)$$

$$= \frac{1}{2}\left(2\eta_{[k]} - 2\eta_{k''})\mathbb{I}\{g_{k''} > 0\}\right)$$

$$\leq \frac{1}{2}\left(1 - 2\eta_{k''})\mathbb{I}\{g_{k''} > 0\}\right) .$$

Consequently,

$$(25) \leq \frac{1}{2\kappa}\mathbf{E}\left[\sum_{k\in[\kappa]}\left(|2\eta_{[k]} - 1|\mathbb{I}\{g_k(\boldsymbol{x})\left(\eta_{[k]} - 1/2\right) < 0\} + |2\eta_{k''} - 1|\mathbb{I}\{g_k''(\boldsymbol{x})\left(\eta_{k''} - 1/2\right) < 0\}\right)\right]$$

$$\leq \frac{m}{2\kappa}\left(\text{Reg}_{\ell_H}(\phi_f)\right) ,$$

thereby concluding the proof. $\square$

# H    Lower bound on surrogate losses

The following theorem shows that the upper bound of Theorem 6.2 cannot be improved by more than a factor of 2.

**Theorem H.1.** *For any convex classification-calibrated loss function $\ell$, and corresponding $\psi$, any $\epsilon > 0$, $0 < \theta \leq 1 - 2t$, and $s \in \mathbb{N}$, $\Gamma'(1/(1-2t))^{\alpha+1}\theta \leq \mathfrak{s} = \frac{s}{m} \leq \min\{1, \Gamma'\theta^{-\alpha}\}$ where $\alpha = -\log_\theta 2$, and $\Gamma' = 2^{-(\alpha+2)}\alpha^\alpha\left(1/(\alpha+1)\right)^{\alpha+1}$, there exists a probability distribution such that Assumption*

*4.2 holds with parameters $\alpha$ and $C_\alpha = 2$, and Assumption 4.1 holds with parameters $s$, $t \in [0, 1/4)$, and a function $f : \mathcal{X} \mapsto \mathbb{R}^m$ such that $Reg_{\ell_H}(\phi_f) = \theta$, $Reg_{\ell_H}(\phi_f) < \mathfrak{s}\,(2\Gamma(1-2t))^{1+\alpha}$, and*

$$Reg_\ell(f) \leq 2\Gamma\mathfrak{s}\left(\frac{Reg_{\ell_H}(\phi_f)}{\mathfrak{s}}\right)^{\frac{\alpha}{\alpha+1}} \psi\left(\frac{1}{2\Gamma}\left(\frac{Reg_{\ell_H}(\phi_f)}{\mathfrak{s}}\right)^{\frac{1}{\alpha+1}}\right) + \epsilon \tag{26}$$

*where $1 \leq \Gamma = 2^{1/(\alpha+1)}(\alpha+1)/\alpha^{\alpha/(\alpha+1)} \leq 3$.*

*Proof.* We consider a distribution whose noisiness is controlled by $0 < \delta \leq 1$. Specifically, let $\mathcal{D}_\delta$ be a distribution such that, $\forall \boldsymbol{x} \in \mathcal{X}, \eta_j(\boldsymbol{x}) = \frac{1}{2}(1+\delta)$ if $j \in [s]$ otherwise $\frac{1}{2}(1-\delta)$. With this distribution,

$$\Pr\left(\bigcup_j \left\{\boldsymbol{X} \,\middle|\, 0 < |2\eta_j(\boldsymbol{X}) - 1| \leq \epsilon\right\}\right) = \begin{cases} 1 & \text{if } \epsilon \geq \delta \\ 0 & \text{o.w} \end{cases}.$$

In this case, $t \leq \frac{1}{2}(1-\delta)$.

This corresponds to constants $\alpha = -\log_\delta 2 = \log_\delta \frac{1}{2}$, $C_\alpha = 2$ in Assumption 4.2. Notice that such $\alpha$ matches $\delta$, in that when $\delta = 0$ we have $\alpha = 0$, and when $\delta = 1$ we have $\alpha = \infty$. Also, constant $C_\alpha$ turns out to be fixed.

Additionally from Lemma F.1, let

$$\Gamma = 2^{1/(\alpha+1)}(\alpha+1)/\alpha^{\alpha/(\alpha+1)} \tag{27}$$

One can easily see that $1 \leq \Gamma' \leq 3$ by separating the two cases $0 < \alpha \leq 1$ and $1 < \alpha$, and taking derivatives.

For the proof of the theorem, we will use the distribution $D_\delta$ with $\delta = \theta$.

We will also choose $f$ such that $f_j(\boldsymbol{x}) < 0$ if $j \in [s]$, and $f_j(\boldsymbol{x}) > 0$, otherwise. Also, observe that $\mathcal{C}_{\eta_j(\boldsymbol{x})}(f_j(\boldsymbol{x})) \leq H^-(f_j(\boldsymbol{x})) + \epsilon$ (see Section E for the definitions of $\mathcal{C}_\eta$ and $H^-$).

Then,

$$R(f) - R^* = \delta = \theta \tag{28}$$

and

$$\begin{aligned} R_{\ell,j}(f_j) - &R^*_{\ell,j} \\ &= \mathcal{C}_{\eta_j(\boldsymbol{x})}(f_j(\boldsymbol{x})) - H(\eta_j(\boldsymbol{x})) \\ &\leq H^-(f_j(\boldsymbol{x})) - H(\eta_j(\boldsymbol{x})) + \epsilon \\ &= \psi(\theta) + \epsilon, \end{aligned}$$

where $R_{\ell,j}$ and $R^*_{\ell,j}$ are the $\ell$-risk and the Bayes $\ell$-risk for label $j$, respectively. Since this holds for all $j \in [m]$, we have

$$R_\ell(f) - R^*_\ell \leq \psi(\theta) + \epsilon. \tag{29}$$

Now, there are conditions that the parameters must satisfy. First, the distribution must satisfy Assumption 4.2 with $\epsilon = \theta$ and $C_\alpha$. We have

$$1 \leq C_\alpha \theta^\alpha$$
$$\Leftrightarrow \left(\frac{1}{2}\right)^{-\log_2 \delta} \leq \theta$$
$$\Leftrightarrow \delta \leq \theta.$$

Since $\theta = \delta$ in our case, this is always satisfied.

Also, we have to show that the following condition is met

$$\frac{1}{2\Gamma}\mathfrak{s}^{-\frac{1}{1+\alpha}}\theta^{\frac{1}{1+\alpha}} \leq 1 - 2t.$$

By plugging in (27),

$$2^{-(\alpha+2)/(\alpha+1)}\alpha^{\alpha/(\alpha+1)}(1/(\alpha+1))\mathfrak{s}^{-1/(\alpha+1)}\theta^{1/(\alpha+1)} \le 1-2t$$

$$\Leftrightarrow 2^{-(\alpha+2)}\alpha^{\alpha}\theta\left(\frac{1}{(\alpha+1)(1-2t)}\right)^{\alpha+1} \le \mathfrak{s} \tag{30}$$

which is the given condition from $\Gamma'(1/(1-2t))^{\alpha+1}\theta \le \mathfrak{s}$ in the statement of the theorem, where $\Gamma' = 2^{-(\alpha+2)}\alpha^{\alpha}\left(1/(\alpha+1)\right)^{\alpha+1}$, and we can see that the LHS is less than 1 when $t < 1/4$.

Further, the following condition is needed at the end:

$$2\Gamma\mathfrak{s}^{\frac{1}{\alpha+1}}\theta^{\frac{\alpha}{\alpha+1}} \le 1 . \tag{31}$$

By plugging in (27), we can write

$$2^{(\alpha+2)/(\alpha+1)}(1/\alpha)^{\alpha/(\alpha+1)}(\alpha+1)\mathfrak{s}^{1/(\alpha+1)}\theta^{\alpha/(\alpha+1)} \le 1$$

$$\Leftrightarrow \mathfrak{s} \le 2^{-(\alpha+2)}\alpha^{\alpha}\theta^{-\alpha}\left(1/(\alpha+1)\right)^{\alpha+1} , \tag{32}$$

which is satisfied from the given condition via $\mathfrak{s} \le \min\{1, \Gamma'\theta^{-\alpha}\}$. One can check that there exists $\mathfrak{s}$ that satisfies both (30) and (32) since $\theta < 1 - 2t$.

We are now in a position to complete the proof. We have

$$R_\ell(f) - R_\ell^*$$
$$\le \psi(R(f) - R^*) + \epsilon \qquad\qquad (\because (29))$$
$$\le \psi\left(\frac{(R(f) - R^*)^{\frac{1}{\alpha+1}}}{2\Gamma\mathfrak{s}^{\frac{1}{\alpha+1}}} \cdot 2\Gamma\mathfrak{s}^{\frac{1}{\alpha+1}}(R(f) - R^*)^{\frac{\alpha}{\alpha+1}}\right) + \epsilon$$
$$\le 2\Gamma\mathfrak{s}^{\frac{1}{\alpha+1}}(R(f) - R^*)^{\frac{\alpha}{\alpha+1}}\psi\left(\frac{(R(f) - R^*)^{\frac{1}{\alpha+1}}}{2\Gamma\mathfrak{s}^{\frac{1}{\alpha+1}}}\right) + \epsilon \qquad (\because (31), \text{Lemma 1 part 1 in [3]}) ,$$

as claimed. $\qquad\qquad\qquad\qquad\qquad\qquad\qquad\qquad\qquad\qquad\qquad\qquad\qquad\qquad\square$