# OpenReview forum: "Regret Bounds for Multilabel Classification in Sparse Label Regimes"
_NeurIPS.cc/2022/Conference — NeurIPS 2022 Accept_

### Official Review · Reviewer_6DSW · 2022-07-10

**Rating:** 8
**Confidence:** 4
**Soundness:** 4 excellent
**Presentation:** 4 excellent
**Contribution:** 4 excellent

**Summary:**

This works studies multi-label classification (MLC) problems. Focusing on sparse label regime (in which the number of active labels is small) and an MLC variant of Tsybakov's low-noise condition,  the paper shows upper and lower regret for two typical performance measures in MLC, namely Hamming Loss and Precision@k, in two settings: non-parametric tackled by K-NN classifiers, and empirical risk minimization (ERM) of surrogate loss functions. It is clearly demonstrated in the paper that sparse label regime and low-noise condition, which are two reasonable conditions for practical considerations as well, can improve the regret bounds.

**Questions:**

In the conclusion section, you said that the analysis can be extended to rank loss, 0/1 loss, and micro/macro F-measures. I don't see how this extension can come easily. As far as I understand, for Hamming Loss and Precision@k, in order to derive Bayes optimal classifiers, it is indeed sufficient to have good estimates of $\eta_j(x)$ for each label $j$. In some sense, the labels are independent of each other. For other complex performance measures, the relations among different labels are crucial in designing Bayes optimal classifiers. Therefore I don't see how the analysis can be extended easily. Can you elaborate on this point?

**Limitations:**

One limitation is that the paper does not have experiments. It would be nice to see some empirical verifications of the results on synthetic and real data sets.

**Strengths And Weaknesses:**

Overall this paper is a very strong piece of theoretical work. It is also well-polished. I really enjoyed reading it. Good work!

**Originality:**
This work is novel. The related work is adequately cited and discussed.

**Quality:**
The work is sound and rigorous.

**Clarity:**
The paper is well-polished. The notations are used consistently. Insights are provided to help readers understand the crux of the theoretical results.

**Significance:**
The paper is an important piece of work in advancing the theoretical guarantees in MLC community. Moreover, it is well-presented, so it could have even more influence.

---
After rebuttal: I have read the other reviews and the authors' responses. My rating for this paper has not changed.

---

> ### Author Response · Authors · 2022-08-02
> **Thanks you for you careful review**
>
> - On extension to rank loss, 0/1 loss and micro/macro F-measures.
>
> Thank you for pointing this out. Indeed, in the process of compressing the paper to the space limit, we shortened this part to a sentence that turned out not to be precise enough. The message we wanted to convey is the following. Based on existing regret bound results for rank loss (e.g., [11]) and micro- and macro-averaged complex performance measures (e.g., [22]), one can get results similar to ours. The crucial point is formulating the right assumptions for those loss functions. The general steps of these analyses will be similar to those contained in our paper. Notice that the Bayes classifiers for these losses are in fact based on marginal quantities (please also check Koyejo et al. 2015, "Consistent Multilabel Classification" -- we have just realized that citation [23] currently in the paper should refer to that 2015 paper instead).
>
> The most challenging case is the 0/1 loss. As observed by the reviewer, this loss is more complex, and one cannot rely on marginal distributions. Nevertheless, we strongly suspect that our current contribution will help obtain the result for this loss function as well.
>
> We will add more explanatory text to the concluding section of the paper.
>
>
> - On the lack of experiments.
>
> Yes, we understand this concern, some experiments are indeed planned. Yet, as it currently stands, this is a theoretical paper, and we would like it to be viewed as such.

---

### Official Review · Reviewer_ZyU8 · 2022-07-11

**Rating:** 7
**Confidence:** 3
**Soundness:** 4 excellent
**Presentation:** 3 good
**Contribution:** 3 good

**Summary:**

This paper focus on the regret bounds for multi-label classification. Under a novel low noise assumption and a sparsity constraint on the posterior possibility $\eta(x)$, the regret upper bounds are provided for surrogate risk minimization w.r.t. Hamming loss and Precision@k. With the sample assumptions and a widely used constraint on smoothness, the high probability bounds are given for a MLC variant of KNN. Regret lower bounds are also provided for KNN to show the tightness of its upper bounds. The given results show that the interplay between low noise assumption and label sparsity can lead to non-trivial improvement on the regret bounds.


**Questions:**

Since $\nu\geq 1$ usually holds, I'm not sure if (5) is better than a common regret bound of $O_p(1/\sqrt{n})$ that is induced without dependence on the low noise and sparsity assumptions.

**Limitations:**

The authors addressed the limitations of the used assumptions and the absence of regret lower bounds for surrogate risk minimization.

**Strengths And Weaknesses:**

Strengths:
1. The assumptions made in this paper are mild and are closely related to the widely accepted ones.

2. The analyses on the MLC variant of KNN are detailed and novel. To the best of my knowledge, there is no work that contributes to the regret bound analysis  of KNN in the regime of MLC.

Weaknesses:

Though the analyses of the MLC variant of KNN are rigorous, the analyzed variant may not be a widely accepted one, which can cause the gap between practice and theory.

---

> ### Author Response · Authors · 2022-08-02
> **Thanks for your careful review**
>
> - On the analyzed MLC variant of kNN.
>
> It is not entirely clear to us which widely accepted variant of kNN is meant by Reviewer ZyU8. We would appreciate if the Reviewer could be more specific here. We may only guess that the Reviewer might perhaps be referring to algorithms like MLkNN [34]. Let us first observe that this algorithm performs additional smoothing by leveraging prior information on the neighborhood labels. Nothing prevents from applying the same smoothing to binary classification and, in fact, MLkNN from [34] is not really a ``multi-label" algorithm, as the inference is done for each label independently. In contrast to that, in our paper we follow the theoretical results obtained for classical kNNs, without any additional smoothing, and the inference needs to leverage global sparsity constraints.
>
> That said, it would be interesting to link the two papers more tightly, but this is outside the scope of our paper.
>
>
> - On the bound in Eq. (5) and dependence on $\nu \geq 1$.
>
> One thing to keep in mind is that in the MLC setting what is relevant is not only the dependence on the sample size $n$, but also the dependence on the sparsity factor $s/m$. This is because in large MLC problems, the total number of classes $m$ can also be greater than the total number of samples $n$. Hence, it is the dependence on both $s/m$ and $Reg_\ell(f)$ (and the interplay between these two quantities) that better explains what is going on in bounds like (5), instead of the sole dependence on $Reg_\ell(f)$. In this sense, for instance, the bound mentioned in line 299 should not be regarded as necessarily worse than the usual bound $n^{-1/2}$.
>
> We will add a comment to the paper to better elucidate this point.

---

### Official Review · Reviewer_gucg · 2022-07-18

**Rating:** 5
**Confidence:** 2
**Soundness:** 3 good
**Presentation:** 3 good
**Contribution:** 3 good

**Summary:**

The paper provides regret bounds for large-scale multilabel classification with Hamming loss and precision-at-k, for both nonparametric (k-nn) and parametric estimation. Efficient scaling of the bounds is achieved by making a sparsity assumption on the labels, which combines a hard sparsity constraint (at most s labels with high probability) with a soft sparsity constraint (all other labels at most probability t < 0.5).

**Questions:**

I'm not convinced that 5.6 is a realistic condition to impose. The number of relevant labels typically varies per data-point, so it seems very strict to demand that the marginals have a steep change exactly between k and k+1 for most of instances. Do you have an argument why this condition would be fulfilled in real data?

l. 235 announces the introduction of two reductions, but the second does not seem to do so.

Assumption 5.1.
should this really be a seminorm, and not an actual norm? A seminorm could just ignore a few of the components (=marginals of labels), so that in these labels there would not need to be any smoothness? Is that deliberate? A norm would also have the nice effect that you need not worry about which norm to use, since this is finite dimensional and any factor could be absorbed by C.

== Supplementary ==
Unfortunately, I have not yet had the time to go through the proofs in the supplementary. When I'm done with that, I will update the review accordingly.

**Limitations:**

the main limitation I see is assumption 5.6, which I doubt is realistic.

**Strengths And Weaknesses:**

The paper provides a step in closing the gap in theory between binary classification and large-scale multilabel classification. The authors manage to incorporate the sparsity property in a way that does not require an unrealistically strict sparsity assumption (i.e. at most s labels, ever). The multilabel-generalization of the Tsybakov-condtition also seems sensible to me, but I doubt that the k-vs-k+1 separation condition needed for the precision-at-k result is realistic.

For the most part, I found it easy enough to follow the writing in the paper. In the regret bounds, the formulas get quite long, so they can become a bit difficult to parse, but I'm not sure if there is anything the authors could have done to improve this.

---

> ### Author Response · Authors · 2022-08-02
> **Thanks for your careful review**
>
> - On Assumption 5.6.
>
> We understand the reviewer's concern here. Regretfully, the margin condition in Assumption 5.6 is generally *inevitable* in order to obtain fast rates for precision@k. To see this, the reviewer should delve into the proof of the lower bound in Theorem 5.10 (Appendix D, l. 622), where it is shown that, for the multi-label problem constructed therein, the margin condition in Assumption 4.2 implies the one in Assumption 5.6 with $\beta = \alpha$. In retrospect, we realized that we formulated Theorem 5.10 in a somewhat misleading way: Assumption 4.2 should be replaced there with Assumption 5.6, and exponent $\alpha$ of Assumption 4.2 should be replaced by exponent $\beta$ of Assumption 5.6. We will make this correction, and better illustrate the necessity of the margin condition in Assumption 5.6 in fast rates for precision@k.
>
> - l. 235 and the two reductions.
>
> The two reductions alluded to in line 235 are those contained in the proofs of Theorem 5.9 and Theorem 5.10. Please see Appendix D for details (l. 614 onward).
>
>
> - On Assumption 5.1: norm vs. semi-norm.
>
> The reviewer is essentially correct. In the paper we end up using this assumption with $||\cdot||$ being the infinity norm, as mentioned in line 503 in Appendix A.
>
> We will better clarify it in the revised version.

---

### Author Response · Authors · 2022-08-02
**General comment**

We thank all reviewers for their useful and insightful comments and glad that our contribution has been well received. We will make appropriate changes to our paper. The responses to the particular comments are given separately to each review.

---

### Meta-Review · Area_Chair_FLEf · 2022-08-26

**Recommendation:** Accept
**Confidence:** Certain

**Metareview:**

This paper studies regret bounds for multi-label classification and derives new upper bounds for surrogate risk minimization under a low-noise assumption. Although there is some concern regarding the conditions, all reviewers support accepting this paper.

**Award:**

No

---

### Decision · Program_Chairs · 2022-09-14

Accept